# LANGUAGE MODELLING WITH PIXELS

**Phillip Rust**[1]   **Jonas F. Lotz**[1,2]   **Emanuele Bugliarello**[1]
**Elizabeth Salesky**[3]   **Miryam de Lhoneux**[5]   **Desmond Elliott**[1,6]
[1]University of Copenhagen    [2]ROCKWOOL Foundation Research Unit
[3]Johns Hopkins University    [5]KU Leuven    [6]Pioneer Centre for AI
p.rust@di.ku.dk

## ABSTRACT

Language models are defined over a finite set of inputs, which creates a *vocabulary bottleneck* when we attempt to scale the number of supported languages. Tackling this bottleneck results in a trade-off between what can be represented in the embedding matrix and computational issues in the output layer. This paper introduces PIXEL, the **Pix**el-based **E**ncoder of **L**anguage, which suffers from neither of these issues. PIXEL is a pretrained language model that renders text as images, making it possible to transfer representations across languages based on orthographic similarity or the co-activation of pixels. PIXEL is trained to reconstruct the pixels of masked patches instead of predicting a distribution over tokens.[1] We pretrain the 86M parameter PIXEL model on the same English data as BERT and evaluate on syntactic and semantic tasks in typologically diverse languages, including various non-Latin scripts. We find that PIXEL substantially outperforms BERT on syntactic and semantic processing tasks on scripts that are not found in the pretraining data, but PIXEL is slightly weaker than BERT when working with Latin scripts. Furthermore, we find that PIXEL is more robust than BERT to orthographic attacks and linguistic code-switching, further confirming the benefits of modelling language with pixels.

## 1   INTRODUCTION

Natural language processing has rapidly progressed in recent years due to a combination of self-supervised representation learning, i.e. pretrained language models (PLMs) like BERT (Devlin et al., 2019), GPT-3 (Brown et al., 2020), and XLM-R (Conneau et al., 2020); large unlabelled datasets; such as C4 (Raffel et al., 2020), The Pile (Gao et al., 2020); and large-scale computing power (Hirschberg & Manning, 2015). Despite this progress, these models only cover a fraction of the world's languages, with large inequalities in performance (Pires et al., 2019; Lauscher et al., 2020), and the majority of languages are falling behind English (Joshi et al., 2020b; Bugliarello et al., 2022). Even within English, these models struggle when tasked with processing noisy inputs (Sun et al., 2020; Eger & Benz, 2020). In this paper, we show how to effectively support *thousands* of written languages in a single model while being robust to variations caused by character-level noise.

Language models typically support a finite vocabulary of categorical inputs, e.g. characters, subwords or even words, and much effort has been devoted to vocabulary construction (Wan, 2022). On one end of the spectrum, a vocabulary over words has three problems: (i) it is not possible to encode out-of-vocabulary words because they lack an entry in a closed vocabulary, e.g. "doxing", (ii) there are too many parameters in the word embedding layer, and relatedly, (iii) the normalising constant for the softmax activation in the output layer is too expensive to compute. On the other end of the spectrum, vocabularies over bytes or characters are much smaller, which leads to increased sequence lengths (Keren et al., 2022). In practice, most current models operate over inputs smaller than words but larger than characters: subword units (Sennrich et al., 2016; Kudo, 2018). Subwords prevent the problem of extremely large embedding and output layers, and support open vocabulary processing. While this is a practical solution in a monolingual context and for some languages like English, dealing with many languages with a variety of scripts will either result in a very large vocabulary or a trade-off over what is represented within a fixed number of subwords (see §5). Taken

---

[1]See Appendix A for reconstructions of this abstract.

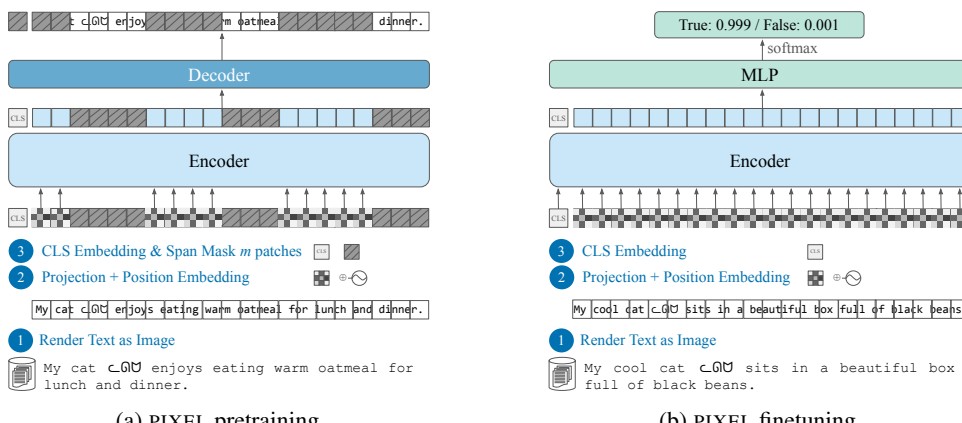

(a) PIXEL pretraining        (b) PIXEL finetuning

Figure 1: Overview of PIXEL's architecture. Following He et al. (2022), we use a masked autoencoder with a ViT architecture and a lightweight decoder for pretraining (left). At finetuning time (right), the decoder is replaced by a task-specific classification head that sits on top of the encoder.

together, given a language model with a finite vocabulary, there is a bottleneck in two locations: at the level of the encoding of the inputs and at the level of estimating the probability distribution over the vocabulary. We call this the *vocabulary bottleneck*. A language model that can handle thousands of languages needs to deal with this problem.

We propose to rethink language modelling as a visual recognition task, removing the need for a finite vocabulary. Our proposal is inspired by Salesky et al. (2021), who showed how to train a machine translation model with "visual text representations" in the encoder instead of subwords. Our **Pix**el-based **E**ncoder of **L**anguage (PIXEL) is built on the Masked Autoencoding Visual Transformer (ViT-MAE; He et al., 2022). ViT-MAE is a Transformer-based encoder-decoder trained to reconstruct the pixels in masked image patches. PIXEL does not have a vocabulary embedding layer; instead, text is rendered as a sequence of fixed-sized patches, which are processed using a Vision Transformer encoder (Dosovitskiy et al., 2021). PIXEL also does not have an expensive output layer when it reconstructs the pixels of the masked patches. In effect, PIXEL provides a solution to the vocabulary bottleneck without needing the prohibitively long sequences of character-based models.

PIXEL is pretrained on the same data as BERT, given our computational resources. This means that it has encountered only ~0.05% non-English text (Blevins & Zettlemoyer, 2022).[2] We evaluate PIXEL on a range of syntactic and semantic tasks in 32 typologically diverse languages across 14 scripts, showing that it can rapidly adapt to new languages and unseen scripts. PIXEL is also evaluated on its ability to handle noisy text caused by orthographic attacks, where pixel-based encoding is a clear improvement over subword-based vocabularies. In lexical code-switching experiments, PIXEL performs on-par with BERT and sometimes outperforms the multilingually pretrained mBERT.

PIXEL is a new type of language model that can theoretically support any language that can be typeset by a modern computer. We make the implementation, the pretrained model including intermediate training checkpoints, and the fine-tuned models freely available for the community.[3]

## 2   APPROACH

The Pixel-based Encoder of Language, PIXEL, consists of three major components: a text renderer, which draws text as an image; an encoder, which encodes the unmasked regions of the image; and a decoder, which reconstructs the masked regions at the pixel level. Figure 1 provides an illustration.

### 2.1   TEXT RENDERER

The key component of PIXEL is a text renderer that takes one or more pieces of text and renders them onto a blank RGB image $x \in \mathbb{R}^{H \times W \times C}$. We set height $H = 16$ and width $W = 8464$ and choose

---

[2]We do not claim that a language model designed to support thousands of languages should be pretrained only on English text. We expect that pretraining on an appropriate choice of another language or multilingually may provide more remarkable results. PIXEL represents an initial effort at smaller scale.

[3]https://github.com/xplip/pixel

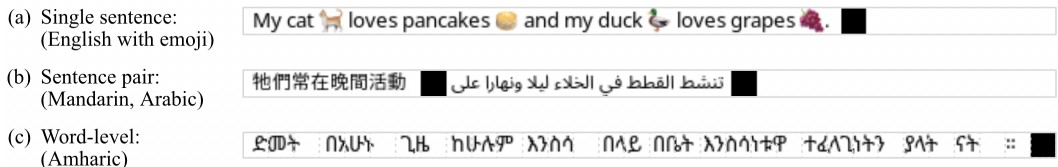

(a) Single sentence: (English with emoji)

(b) Sentence pair: (Mandarin, Arabic)

(c) Word-level: (Amharic)

Figure 2: Illustrative examples of our rendered text. PIXEL natively supports most writing systems, colour emoji (a), and complex text layouts such as right-to-left writing and ligatures (b). Black patches serve as separators and end-of-sequence markers. Blank patches to the right of the end-of-sequence marker are treated as sequence padding. For word-level tasks, horizontal spacing can be added between words (c) so that every patch can be assigned to exactly one word (dotted lines indicate patch boundaries for demonstration).

$C = 3$ RGB input channels, which is equivalent to a square colour image with a $368 \times 368$ resolution and corresponds to a sequence of 529 image patches of size $16 \times 16$ pixels.[4] Figure 2 shows examples of text inputs rendered by the text renderer. The renderer supports (a) colour emoji and hieroglyphs scripts, (b) left-to-right and right-to-left writing systems, and (c) text that requires ligatures. Analogous to BERT, a sequence can either contain a single paragraph of text or a text pair; we use black $16 \times 16$ patches to serve as separators and end-of-sequence (EOS) markers. Blank (white) patches after the end-of-sequence marker are treated as padding by PIXEL, where no attention scores or losses are computed. Sequences longer than the maximum length are either truncated or split into multiple sequences. Further technical details about the renderer are provided in Appendix D.

## 2.2 ARCHITECTURE

PIXEL-base is a 112M parameter ViT-MAE architecture (He et al., 2022) with a 12-layer ViT encoder (Dosovitskiy et al., 2021) and an 8-layer Transformer decoder (Vaswani et al., 2017). The encoder has 86M parameters and the decoder has 26M parameters, respectively. The 8-layer decoder is not used for downstream tasks. We give an overview of the architecture below, with more details in Appendix E. We did not train larger PIXEL variants for lack of computational resources.

**Patch Embeddings** The images produced by the text renderer (§2.1) are patch-wise linearly projected to obtain a sequence of patch embeddings with a $16 \times 16$ pixel resolution, to which fixed sinusoidal position embeddings are added.[5]

**Patch Span Masking** Instead of the random masking procedure used in ViT-MAE or block-wise masking in BEiT (Bao et al., 2022), PIXEL uses span masking with a 25% masking ratio as outlined in Algorithm 1, which masks spans of up to $S = 6$ consecutive image patches with a dynamic number of unmasked patches left between them. The idea behind the span masking approach, inspired by T5 (Raffel et al., 2020) and SpanBERT (Joshi et al., 2020a), is that it masks more meaningful units of text (full words or phrases) than random masking where the model more often has to fill in (parts of) individual characters, thereby encouraging PIXEL to model a higher level of abstraction. In practice, span masking was slightly more effective than random masking in early prototypes of PIXEL.

---

**Algorithm 1** PIXEL Span Masking

**Input:** #Image patches $N$, masking ratio $R$, maximum masked span length $S$, span length cumulative weights $W = \{w_1, \ldots, w_S\}$
**Output:** Masked patches $\mathcal{M}$
$\mathcal{M} \leftarrow \emptyset$
**repeat**
  $s \leftarrow \text{randchoice}(\{1, \ldots, S\}, W)$
  $l \leftarrow \text{randint}(0, \max(0, N - s))$
  $r \leftarrow l + s$
  **if** $\mathcal{M} \cap \{l - s, \ldots, l - 1\} = \emptyset$ **and**
    $\mathcal{M} \cap \{r + 1, \ldots, r + s\} = \emptyset$ **then**
    $\mathcal{M} \leftarrow \mathcal{M} \cup \{l, \ldots, r\}$
  **end if**
**until** $|\mathcal{M}| > R \cdot N$ **return** $\mathcal{M}$

---

This effect may be less noticeable at higher masking ratios (such as the 75% used in ViT-MAE), when random masking would more often masks consecutive patches. We found 25% masking ratio to work well for PIXEL-base, which is in line with recent findings for BERT-type models of similar size (Wettig et al., 2022). We mask spans of $s \in \{1, 2, 3, 4\}$ patches in length, each with 20% probability, and spans of $s \in \{5, 6\}$ patches with 10% probability each, so $\mathbb{E}(s) = 3.1$.

---

[4]We chose a sequence length of 529 so that the memory requirements at maximum length are approx. equal to those of BERT. Forward and backward passes of the transformer layers at equal length are also equally fast.

[5]This is a fast operation that does not require the large text embedding layer found in subword-based models, saving parameters which could in theory be re-allocated to the self-attention stack. We refer to Xue et al. (2022) for a discussion regarding benefits and drawbacks of re-allocation of embedding layer weights.

**Encoder**  Following ViT-MAE (He et al., 2022), the PIXEL encoder only processes unmasked patches (i.e., $\approx 396$ "visible" patches at 25% masking) rather than on a sequence including mask tokens, which not only reduces memory requirements and increases training speed, but also has the advantage of not creating a mismatch between pretraining and finetuning. This mismatch would occur when training the encoder with inserted mask tokens because they are not inserted during finetuning (He et al., 2022). We also prepend the special CLS embedding to the unmasked patches.[6] The resulting CLS and unmasked patches are processed by a 12-layer Transformer encoder to produce a sequence of encoder output representations.

**Decoder**  The PIXEL decoder first projects the encoder outputs into the same space as the decoder model hidden size. It then inserts learnable mask embeddings at the masked positions; these are what PIXEL tries to reconstruct at the pixel level. Fixed sinusoidal position embeddings (Vaswani et al., 2017) are added to inject order information. After processing this sequence via 8 Transformer layers, a linear projection yields patch logits. Note that the decoder does not have to compute an expensive softmax over a subword vocabulary and circumvents the question of whether to tie the subword embedding weights. PIXEL is trained with a normalised mean squared error (MSE) pixel reconstruction loss measuring the discrepancy between normalised target image patches and reconstructed patches. This loss is only computed for *masked, non-blank (text)* patches.

## 2.3 PRETRAINING

PIXEL-base is pretrained on a rendered version of the English Wikipedia and the Bookcorpus (Zhu et al., 2015), which is roughly equivalent to the BERT pretraining data.[7] For better compute efficiency, we concatenate paragraphs until the maximum sequence length is reached, albeit not across document and book boundaries. Wikipedia has 2B words rendered into 11.4M examples and the Bookcorpus has 1.1B words rendered into 5.4M examples; in total ~3.1B words (BERT used 3.3B) rendered into 16.8M examples.[8] PIXEL is pretrained for 1M steps with batch size 256 (i.e. ~16 epochs) using the AdamW optimizer (Kingma & Ba, 2015; Loshchilov & Hutter, 2019) with a linear warmup over the first 50k steps to a peak learning rate of $1.5e{-}4$ and a cosine decay to a minimum learning rate of $1e{-}5$. Pretraining took 8 days on $8\times40$GB Nvidia A100 GPUs. We show the loss curve and additional pretraining details in Appendix E. We stored PIXEL checkpoints every 10k steps and make them available alongside the fully trained model on the HuggingFace Hub (Wolf et al., 2020),which we hope will be useful to analyze training dynamics of PIXEL models (Sellam et al., 2022). Figure 5 in Appendix B shows, for three unseen examples, how PIXEL learns to model language over the course of pretraining.

## 2.4 FINETUNING

PIXEL can be finetuned for downstream NLP tasks in a similar fashion to BERT-like encoders by simply replacing the PIXEL decoder with a suitable classification head. By truncating or interpolating the sinusoidal position embeddings, we can finetune with sequences shorter or longer than 529 patches, respectively. The latter, in particular, is common in computer vision applications to finetune on higher resolution images (Touvron et al., 2019; Kolesnikov et al., 2020; Dosovitskiy et al., 2021; He et al., 2022). For most common NLP tasks, we can typically finetune with sequences shorter than 529 to accelerate training while retaining performance. To demonstrate that PIXEL supports a variety of downstream tasks, we conduct finetuning experiments in four settings as follows:

**Word Classification**  For word-level tasks like part-of-speech (POS) tagging and named entity recognition (NER), we render each word at the start of a new image patch so that we can create a bijective mapping between words and patches (see Figure 2 for an example).[9] To finetune PIXEL on these images, we add a linear classifier with dropout. We assign the label of a word only to its first corresponding image patch and compute a cross-entropy loss with softmax.

**Dependency Parsing**  For dependency parsing, we render text as above but obtain word-level representations by mean pooling over all corresponding image patches of a word and employ a biaffine parsing head (Dozat & Manning, 2017), following the implementation from Glavaš & Vulić (2021).

---

[6]In pretraining, no loss is computed for the CLS embedding but it can be used for finetuning.

[7]We use a similar Wikipedia dump Devlin et al. (2019) used for BERT (February 1, 2018) and a slightly newer version of the Bookcorpus available at `https://huggingface.co/datasets/bookcorpusopen`.

[8]This rendering is quite compact; see Appendix D.

[9]This particular formulation assumes that word boundaries are available. We note that subword-based and character-based models also make this assumption. For further discussion on the implications, see Appendix F.

**Sequence Classification** For sequence-level tasks, e.g. in GLUE (Wang et al., 2018), we render text as in pretraining. For sentence-pair tasks like natural language inference (NLI) we separate the sentences with a black patch. We finetune with different strategies, including training a classifier on top of (1) the CLS embedding, (2) the mean-pooled or max-pooled representations of all patches, (3) a multi-head attention block. Although we did not notice significant performance differences between them in our experiments, we mainly used option (1), which is exactly the same as in BERT, and (2), which has been shown to work well for image classification (Liang et al., 2022).

**Extractive Question Answering (QA)** For extractive QA datasets like SQuAD (Rajpurkar et al., 2016), we render the question and context like in sequence-pair tasks above and, same as Devlin et al. (2019), use a sliding window approach to extract answers for examples exceeding the maximum sequence length. We use a linear classifier to predict the start and end patches of the span containing the answer. Appendix D explains how we obtain the mapping between characters and rendered text.

## 3 EXPERIMENTS

We finetune PIXEL on common NLP tasks and evaluate its syntactic and semantic processing capabilities in English, as well as its adaptability to unseen languages. Table 8 (Appendix F) describes the languages used in these experiments, and our language and data selection is also motivated below.

### 3.1 TASKS AND LANGUAGES

**Syntactic Tasks** We evaluate PIXEL on part-of-speech (POS) tagging and dependency parsing using data from Universal Dependencies v2.10 treebanks (Nivre et al., 2020; Zeman et al., 2022) for a set of typologically diverse languages that captures a large variety of unseen scripts[10]: Arabic (ARA), Coptic (COP), English (ENG), Hindi (HIN), Japanese (JPN), Korean (KOR), Tamil (TAM), Vietnamese (VIE), Chinese (ZHO).[11] We compare how well PIXEL transfers to these languages compared to BERT. Note that BERT does not support all of these writing systems. However, both models have been trained on the same data. This comparison allows us to gauge the extent to which PIXEL can overcome the script barrier and vocabulary bottleneck of subword-based models.

**Semantic Tasks** We evaluate both monolingual (ENG) and cross-lingual *word-level* understanding on MasakhaNER (Adelani et al., 2021), a named entity recognition (NER) benchmark for 10 African languages (AMH, HAU, IBO, KIN, LUG, LUO, PCM, SWA, WOL, YOR), which also includes a copy of the ConLL-2003 dataset (ENG; Tjong Kim Sang & De Meulder, 2003). For monolingual ENG *sentence-level* understanding we rely on GLUE (Wang et al., 2018) and SQuAD (Rajpurkar et al., 2016). Finally, we evaluate cross-lingual sentence-level understanding on TyDiQA-GoldP (Clark et al., 2020) in the *in-language multitask* setting where we train on the combined gold data in all 9 target languages (ARA, BEN, ENG, FIN, IND, KOR, RUS, SWA, TEL) at once, and on two additional larger monolingual extractive question answering (QA) corpora: KorQuAD 1.0 (KOR; Lim et al., 2019) and JaQuAD (JPN; So et al., 2022).

### 3.2 BASELINES AND FINETUNING PROTOCOLS

We compare results to BERT-base which is trained on the same data.[12] We do not compare to newer monolingual English models like ROBERTA (Liu et al., 2019), T5 (Raffel et al., 2020) or DEBERTA (He et al., 2021b;a) because these models have been pretrained longer on much larger corpora.[13] Likewise, we do not compare against models trained on massively multilingual corpora. However, to contextualise the performance of PIXEL in cross-lingual settings, we report results for MBERT and, if results are available, for CANINE (Clark et al., 2022). For BERT, we use the standard finetuning protocols used by Devlin et al. (2019) and the same biaffine classifier for parsing as for PIXEL. We list finetuning details for all tasks in Appendix F.

### 3.3 RESULTS

**Syntactic Tasks** We present results for POS tagging and dependency parsing in Table 1. While BERT is slightly better than PIXEL in the monolingual setting (ENG), PIXEL clearly outperforms

---

[10]By unseen, we mean not present in the pretraining data.

[11]Table 10 in Appendix F gives an overview of the treebanks we use.

[12]We use BERT weights from `https://huggingface.co/bert-base-cased`.

[13]We do not intend to claim state-of-the-art performance, but to demonstrate that PIXEL can overcome the vocabulary bottleneck and to provide a starting point for further research on pixel-based encoding of language.

| | $|\theta|$ | ENG | ARA | COP | HIN | JPN | KOR | TAM | VIE | ZHO |
|---|---|---|---|---|---|---|---|---|---|---|
| | | | | | *POS Tagging (Accuracy)* | | | | | |
| BERT | 110M | **97.2** | 95.4 | 26.5 | 86.4 | 87.9 | 60.0 | 45.4 | 84.5 | 58.6 |
| PIXEL | 86M | 96.7 | **95.7** | **96.0** | **96.3** | **97.2** | **94.2** | **81.0** | **85.7** | **92.8** |
| | | | | | *Dependency Parsing (LAS)* | | | | | |
| BERT | 110M | **90.6** | **77.7** | 13.0 | 75.9 | 73.8 | 30.2 | 15.2 | 49.4 | 28.8 |
| PIXEL | 86M | 88.7 | 77.3 | **83.5** | **89.2** | **90.7** | **78.5** | **52.6** | **50.5** | **73.7** |

| | [UNK]% | Fertility |
|---|---|---|
| ENG | 0 | 1.2 |
| ARA | 1.8 | 3.7 |
| COP | 93.6 | 1.0 |
| HIN | 32.6 | 2.7 |
| JPN | 45.5 | 1.5 |
| KOR | 84.7 | 1.0 |
| TAM | 82.3 | 1.3 |
| VIE | 4.5 | 2.5 |
| ZHO | 73.2 | 1.5 |

Table 1: Results for PIXEL and BERT finetuned for POS tagging and dependency parsing on various Universal Dependencies treebanks. We report test set results averaged over 5 runs each. $|\theta|$ denotes the number of model parameters. The table on the right shows BERT's proportion of [UNK]s as a measure of (inverse) vocabulary coverage and fertility (i.e., number of subwords per tokenized word; Ács, 2019; Rust et al., 2021) as a measure of over-segmentation in respective UD treebanks.

| | #L | $|\theta|$ | ENG | AMH | HAU | IBO | KIN | LUG | LUO | PCM | SWA | WOL | YOR |
|---|---|---|---|---|---|---|---|---|---|---|---|---|---|
| MBERT* | 104 | 179M | 92.2 | 0 | 87.3 | 85.3 | 72.6 | 79.3 | 73.5 | 86.4 | 87.5 | 62.2 | 80.0 |
| CANINE-C + n-gram* | 104 | 167M | 89.8 | 50.0 | 88.0 | 85.0 | 72.8 | 79.6 | 74.2 | 88.7 | 83.7 | 66.5 | 79.1 |
| CANINE-C* | 104 | 127M | 79.8 | 44.6 | 76.1 | 75.6 | 58.3 | 69.4 | 63.4 | 66.6 | 72.7 | 60.7 | 67.9 |
| BERT | 1 | 110M | **92.9** | 0 | **86.6** | **83.5** | **72.0** | **78.4** | **73.2** | **87.0** | **83.3** | **62.2** | **73.8** |
| PIXEL | 1 | 86M | 89.5 | **47.7** | 82.4 | 79.9 | 64.2 | 76.5 | 66.6 | 78.7 | 79.8 | 59.7 | 70.7 |

Table 2: Results for PIXEL and BERT finetuned for NER on MasakhaNER. We report test set $F_1$ scores averaged over 5 runs each. BERT outperforms PIXEL in all of the languages that use Latin script, whereas PIXEL does better on AMH, whose script is not covered by BERT's vocabulary. The performance gap is smaller for languages heavier in diacritics, e.g. YOR. It is larger for languages closer to English such as Naija Pidgin (PCM), an English-based creole. #L denotes the number of pretraining languages and * indicates results taken from Clark et al. (2022) for additional context.

BERT in the remaining languages. On the lower end, the accuracy gap in favor of PIXEL in ARA and VIE, both languages covered by BERT's vocabulary, is relatively small (∼1%). On the higher end, in COP, where BERT has an out-of-vocabulary ([UNK]) token ratio of 93%, the gap is ∼70% for both tasks. There is a strong correlation[14] between the proportion of [UNK]s (shown in Table 1 on the right) and the performance gap, which shows that PIXEL overcomes BERT's vocabulary bottleneck. These results are further analysed in Appendix I.

**Semantic Tasks** We present results for NER in Table 2, for GLUE in Table 3, for QA in Table 4. We also conduct experiments on XNLI in the *translate-train-all* setting which we present in Table 16 in Appendix I, for brevity. We find that BERT consistently achieves higher performance than PIXEL in its pretraining language ENG. Likewise, it often outperforms on languages using the Latin writing system; for instance in NER where all languages besides AMH use Latin script, in QA for FIN, IND and SWA. Although BERT has more trainable parameters, this finding indicates that a PIXEL model pretrained for the same number of steps as BERT is slightly worse at semantic tasks, and it may require longer pretraining or an additional inductive bias to close the performance gap. Similarly, character-based models also tend to underperform subword-based models on NER (Keren et al., 2022), here seen by the CANINE-C results. Since the addition of n-gram embeddings improves the performance of CANINE-C, likely due to boosting entity memorisation capabilities (Clark et al., 2022), we hypothesize that PIXEL may benefit from equivalent enhancements.

For languages where BERT only partially covers the script, such as KOR, JPN and TEL in QA, PIXEL consistently outperforms BERT, sometimes by large amounts (e.g. , +63 $F_1$ points better on Ko-rQuAD). In the extreme case where BERT has no coverage of the script whatsoever, seen in NER for AMH, BERT fails completely (0 $F_1$) while PIXEL outperforms the larger, multilingually trained CANINE and performs competitively with its n-gram variant. In other words, PIXEL also overcomes the vocabulary bottleneck of subword-based PLMs in semantics-driven tasks. Note that although BERT was trained on English, its vocabulary has a high coverage of the Arabic script, explaining its good performance in ARA and URD.[15]

---

[14] Pearson correlation $r = 0.9$, $p < 0.001$ for POS tagging, $r = 0.95$, $p < 0.0001$ for dependency parsing.

| | $|\theta|$ | MNLI-M/MM 393k | QQP 364k | QNLI 105k | SST-2 67k | COLA 8.6k | STS-B 5.8k | MRPC 3.7k | RTE 2.5k | WNLI 635 | AVG |
|---|---|---|---|---|---|---|---|---|---|---|---|
| BERT | 110M | **84.0 / 84.2** | **87.6** | **91.0** | **92.6** | **60.3** | **88.8** | **90.2** | **69.5** | 51.8 | **80.0** |
| PIXEL | 86M | 78.1 / 78.9 | 84.5 | 87.8 | 89.6 | 38.4 | 81.1 | 88.2 | 60.5 | **53.8** | 74.1 |

Table 3: Results for PIXEL and BERT finetuned on GLUE. We report *validation* set performance averaged over 5 runs. The metrics are $F_1$ score for QQP and MRPC, Matthew's correlation for COLA, Spearman's $\rho$ for STS-B, and accuracy for the remaining datasets. PIXEL achieves non-trivial performance scores on GLUE, indicating *pixel-based encoders can learn higher-level semantic tasks*, but performs worse overall than BERT, so it may require (a) more pretraining steps than subword-tokenized PLMs or (b) additional inductive bias to acquire the same level of monolingual abstraction.

| | #L | $|\theta|$ | | | | | TyDiQA-GoldP | | | | | | SQuAD | KorQuAD | JaQuAD |
|---|---|---|---|---|---|---|---|---|---|---|---|---|---|---|---|
| | | | ENG | ARA | BEN | FIN | IND | KOR | RUS | SWA | TEL | **AVG** | ENG | KOR | JPN |
| MBERT | 104 | 179M | 75.6 | 78.1 | 74.7 | 75.5 | 84.3 | 64.8 | 74.9 | 83.1 | 81.6 | 77.1 | 88.6 | 90.0 | 76.4 |
| BERT | 1 | 110M | **68.5** | **58.0** | **43.2** | **58.3** | **67.1** | 12.4 | **53.2** | **71.3** | 48.2 | 51.5 | **88.2** | 14.9 | 28.8 |
| PIXEL | 1 | 86M | 59.6 | 57.3 | 36.3 | 57.1 | 63.6 | **26.1** | 50.5 | 65.9 | **61.7** | **52.3** | 81.4 | **78.0** | **34.1** |

Table 4: Results for PIXEL and BERT finetuned on extractive QA datasets. We report validation set $F_1$ scores averaged over 5 runs each. Average (AVG) scores for TyDiQA-GoldP exclude ENG as customary (Clark et al., 2020). While BERT clearly outperforms PIXEL in ENG, PIXEL is much better in KOR, TEL, and JPN—a consequence of the vocabulary bottleneck in BERT—thereby gaining an edge on average. In some languages, answer span extraction adversely affects results (see §3.3).

While the same may apply to languages like BEN and RUS in QA, where one may otherwise expect PIXEL to outperform BERT, there is an external factor at play; in the standard QA task formulation used by BERT, answer spans are extracted by predicting start and end tokens. We adopt this procedure in PIXEL for simplicity. However, an image patch will often overlap two words at variable positions, so the answer may actually start or end mid-patch. By only predicting on a full-patch level, and extracting the entire content of the patch, PIXEL will sometimes extract leading and trailing characters that should not be part of the answer, which degrades the $F_1$ score—even though the model may have correctly identified the span. Languages not using whitespace to delimit words are particularly affected, which also explains why PIXEL is only slightly better than BERT in JPN.

Generally, and in particular when transferring to unseen scripts, we find that PIXEL performs best when finetuning on larger corpora. An example of this behaviour can be seen in QA, where PIXEL performs significantly better on KorQuAD (60k examples) than the KOR subset of TyDi (1.6k examples). While large corpora may often not be available when dealing with unseen scripts, we hypothesize that multilingual pretraining will alleviate the need for long finetuning, while potentially being even more conducive to *positive transfer* (Conneau et al., 2020; Chau et al., 2020; Pfeiffer et al., 2021) by not being vocabulary-bottlenecked.

## 4 ROBUSTNESS TO ORTHOGRAPHIC ATTACKS AND CODE-SWITCHING

Informal text, commonly found on social media, often contains orthographic noise such as typos and other variations (Baldwin et al., 2015; van Esch et al., 2019; Caswell et al., 2020). Previous work has demonstrated the vulnerability of pretrained language models to character-level adversarial attacks and noise (Sun et al., 2020; Eger & Benz, 2020), with text normalization typically required to maintain performance (Pruthi et al., 2019; Keller et al., 2021). To evaluate PIXEL's robustness to textual noise and variation and inspired by the robustness tests of Salesky et al. (2021), we experiment with the *Zeroé* benchmark (Eger & Benz, 2020; Keller et al., 2021) which covers a variety of low-level orthographic attacks as illustrated in Table 13. We replace their version of visual attacks with the Unicode Technical Standard #39 set of visually-confusable characters.[16] We apply *Zeroé* attacks during finetuning and evaluation of two English downstream tasks, POS tagging and NLI (Bowman et al., 2015), where we expect models to rely on different levels of abstraction.

---

[15] Arabic is lexically sparse (Antoun et al., 2020; Al-Sallab et al., 2017), so the characters can be covered in the vocabulary. However, it is morphologically complex, which leads to over-segmentation, as the fertility of 3.7 in Table 1 shows. This over-segmentation is not necessarily problematic in our selection of tasks (Keren et al., 2022), e.g. due to the sliding window in QA, but can be a disadvantage in others (Rust et al., 2021).

[16] https://util.unicode.org/UnicodeJsps/confusables.jsp

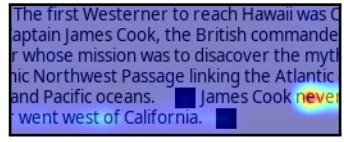 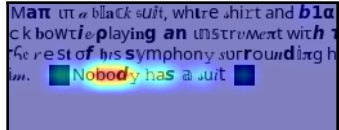 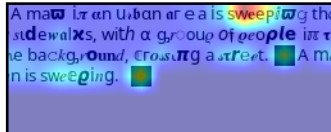

(a) 0%, *contradiction*   (b) 80%, *contradiction*   (c) 80%, *entailment*

Figure 3: Visual explanations of correct PIXEL predictions (for classes *contradiction* and *entailment*) for NLI examples with 0% and 80% CONFUSABLE substitutions using method by Chefer et al. (2021), providing qualitative evidence for PIXEL's robustness to character-level noise and the interpretability of its predictions. Red heatmap regions represent high relevancy.

Figures 8 and 9 in Appendix G compare PIXEL and BERT across three levels of token-level noise for POS tagging and NLI. There is little impact on POS tagging performance with either model from most low-level attacks, with the exception of visually-confusable character substitutions (CONFUSABLE); here PIXEL expectedly maintains performance above 92% as it generalizes across orthographic similarities but BERT drops to 38%. For NLI, both models are negatively affected, but PIXEL exhibits less degradation than BERT with higher proportions of noise, with the impact varying across the types of attacks which each affect subword tokenization differently. Figure 3 shows relevancy heatmaps (Chefer et al., 2021) for SNLI predictions made with and without CONFUSABLE substitutions. The heatmaps are similarly clear with and without noise, providing qualitative evidence that PIXEL is indeed robust to the noise. The illustrated robustness may be dependent upon finetuning, however; we find that PIXEL can struggle in zero-shot applications when text is rendered differently from observed during pretraining (see Appendix D on using different fonts). Future work could explore the impact of data augmentation during pretraining on PIXEL's robustness and ability to transfer across scripts. Furthermore, it would be interesting to investigate how the choice of font influences the search space during reconstruction of masked patches (Bland et al., 2022).

In addition to robustness to orthographic noise, dealing with character-level substitutions is important for effectively modelling different morphological forms. There are also many types of higher-level token, phrase or sequence-level variations such as code-switching—when a speaker alternates between two or more languages in the same utterance, while being grammatically consistent in each language (Joshi, 1982)—or the lexical substitutions in social media text. We evaluate PIXEL on the LinCE benchmark (Aguilar et al., 2020), which includes core tasks and downstream applications for linguistic code-switching. PIXEL is fine-tuned on POS Tagging and NER in Spanish-English, Hindi-English and Modern Standard Arabic-Egyptian Arabic. Table 5 shows that PIXEL and BERT perform similarly on SPA-ENG tasks, with BERT outperforming PIXEL on NER for (romanised) HIN-ENG. On the other tasks, PIXEL performs better than BERT and even outperforms MBERT on HIN-ENG POS tagging. The gap between MBERT and PIXEL is larger on Arabic scripts, which were extensively seen by MBERT during pretraining.

|  | POS Tagging | | Named Entity Recognition | | |
|---|---|---|---|---|---|
|  | SPA-ENG | HIN-ENG | SPA-ENG | HIN-ENG | MSA-EA |
| MBERT | 97.1 | 86.3 | 64.0 | 72.6 | 65.4 |
| BERT | **96.9** | 87.0 | **61.1** | **74.5** | 59.4 |
| PIXEL | 96.8 | **88.2** | 61.0 | 73.0 | **63.7** |

Table 5: Code-switching results on LINCE.

## 5   RELATED WORK

The question of vocabulary construction is an open problem in NLP, especially in a multilingual context.[17] The most widely used language models, e.g. BERT, RoBERTa, T5, GPT-2 *inter alia*, rely on different tokenizers, such as WordPiece (Devlin et al., 2019), Byte-Pair Encoding (BPE; Sennrich et al., 2016) and Unigram LM (Kudo, 2018). There is an established ecosystem around subword tokenizers, such as the SentencePiece (Kudo & Richardson, 2018) and HuggingFace Tokenizers.

In a monolingual context and for some languages like English, vocabularies of subwords are a good tradeoff between vocabularies of characters and vocabularies of words. When representing a large number of languages in multilingual PLMs like mBERT and XLM-R, adequately representing the vocabulary of each individual language would be computationally prohibitive. The tokenization then becomes a bottleneck when trying to scale up to a large number of languages (Conneau et al., 2020; Rust et al., 2021), which manifests itself in degraded cross-lingual performance to languages and

---

[17]See Mielke et al. (2021) for a recent, comprehensive survey on open-vocabulary modeling and tokenization.

language families that are underrepresented in the data used for training multilingual PLMs. There are large inequalities in the performance of these models across typologically diverse languages (Wu & Dredze, 2020; Lauscher et al., 2020). This issue is further exacerbated by tokenizations out-of-the-box not being compatible across languages (Maronikolakis et al., 2021). Language imbalance and poor character coverage in the vocabulary can also decrease downstream performance (Zhang et al., 2022). To some extent, these problems can be attenuated through techniques such as subword mapping (Vernikos & Popescu-Belis, 2021), transliteration (Moosa et al., 2022), leveraging lexical overlap (Patil et al., 2022), vocabulary clustering and reallocation (Chung et al., 2020), continued or language-adaptive pretraining (Ebrahimi & Kann, 2021), adaptation via bilingual lexica (Wang et al., 2022), and embedding matrix adaptation (Artetxe et al., 2020). However, these are post-hoc workarounds to expand model vocabularies after training. They do not provide a direct solution to the vocabulary bottleneck problem.

Some subword-based algorithms can also produce undesirable segmentations for morphologically rich languages (Klein & Tsarfaty, 2020; Amrhein & Sennrich, 2021), so dedicated morphologically-aware tokenizers have been developed (e.g. Smit et al. (2014)), but this process often requires expert-level knowledge and may only work for individual languages.

Due to the limitations of subword vocabularies in multilingual language modelling, some works have used vocabularies over characters (Lee et al., 2017; Ma et al., 2020, *inter alia*) or bytes (Wang et al., 2020; Wei et al., 2021). These provide benefits over purely subword-based models in terms of robustness and most of them are readily applicable in a multilingual context,[18] but they typically come at the cost of increased sequence lengths or latency. Also, such models cannot exploit orthographic similarities between characters across and within scripts and do not account for the fact that meaning of language may be carried visually such as in writing systems that are (partially) logographic like Chinese, in ancient hieroglyphs, or when using emoji.

Finally, some works have developed pixel-based approaches. Broscheit (2018) embedded images of Chinese glyphs but still relied on a fixed vocabulary. Wu et al. (2019) combined character-level images and embeddings for a variety of Chinese tasks. Radford et al. (2021) trained a linear probe for CLIP, which also incorporates a tokenizer, on a rendered version of SST-2 (Socher et al., 2013). Other works have trained pixel-based models that removed the need for a fixed vocabulary: Sun et al. (2019) trained a convolutional sentiment classifier on pixels. Mansimov et al. (2020) used images of text for in-image MT. Salesky et al. (2021) employed a convolutional embedder for a Transformer-based MT system with a subword-based decoder. Our method differs from these in that it provides a general-purpose language encoder that completely removes the need for a vocabulary.

## 6  CONCLUSION

This paper introduced PIXEL, a pretrained language model that renders text as images, which allows it to represent any written language that can be typeset using its text renderer. PIXEL was pretrained on the predominantly English Wikipedia and Bookcorpus datasets, and evaluated on part-of-speech tagging, dependency parsing, question answering, and language understanding tasks. The results demonstrate that PIXEL readily transfers to unseen scripts, as shown by its performance on 14 scripts across 32 languages. PIXEL currently lags behind BERT when processing languages with a Latin script, including English; however, PIXEL is more robust than BERT against low-level orthographic attacks and performs competitively to BERT and mBERT on linguistic code-switching tasks. Overall, these results show that pixel-based representations are a strong backbone for cross-lingual and cross-script transfer learning. The limitations of this work are discussed in Appendix J.

In future work, we will investigate inductive biases and additional objectives that can better capture long-range dependencies in PIXEL models. We hope that this will help overcome the limits of PIXEL in semantic processing. We also plan to pretrain PIXEL on multilingual text with a view to further improving its cross-script and cross-lingual abilities. This will also allow us to more fairly compare pixel-based models against larger subword-based and tokenization-free *multilingual* models. Finally, we will also develop new rendering and finetuning formulations that are better tailored to pixel-based models, e.g. for improving downstream question answering.

---

[18]Character-aware models are not directly applicable to languages that do not use whitespace to delimit sentences (Tay et al., 2021), for example.

## ACKNOWLEDGMENTS

We thank Ákos Kádár, Barbara Plank, and Kris Cao for their comments on an earlier draft. We also thank Davide Rigoni, Rita Ramos, Stella Frank, and members of the CoAStaL and LAMP groups for discussions. Miryam de Lhoneux is funded by the Swedish Research Council (grant 2020-00437). Phillip Rust is funded by the Novo Nordisk Foundation (grant NNF 20SA0066568). Jonas F. Lotz is funded by the ROCKWOOL Foundation (grant 1242). ▪ Emanuele Bugliarello is supported by funding from the European Union's Horizon 2020 research and innovation programme under the Marie Skłodowska-Curie grant agreement No 801199. Elizabeth Salesky is supported by the Apple Scholars in AI/ML fellowship. Desmond Elliott is partially supported by the Innovation Foundation (grant 0176-00013B) and the Novo Nordisk Foundation (grant NNF 20SA0066568). This work was supported by a research grant (VIL53122) from VILLUM FONDEN. The computing power was generously supported by EuroHPC grants 2010PA5869, 2021D02-068, and 2021D05-141, and with Cloud TPUs from Google's TPU Research Cloud (TRC).

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

# A   ABSTRACT RECONSTRUCTIONS

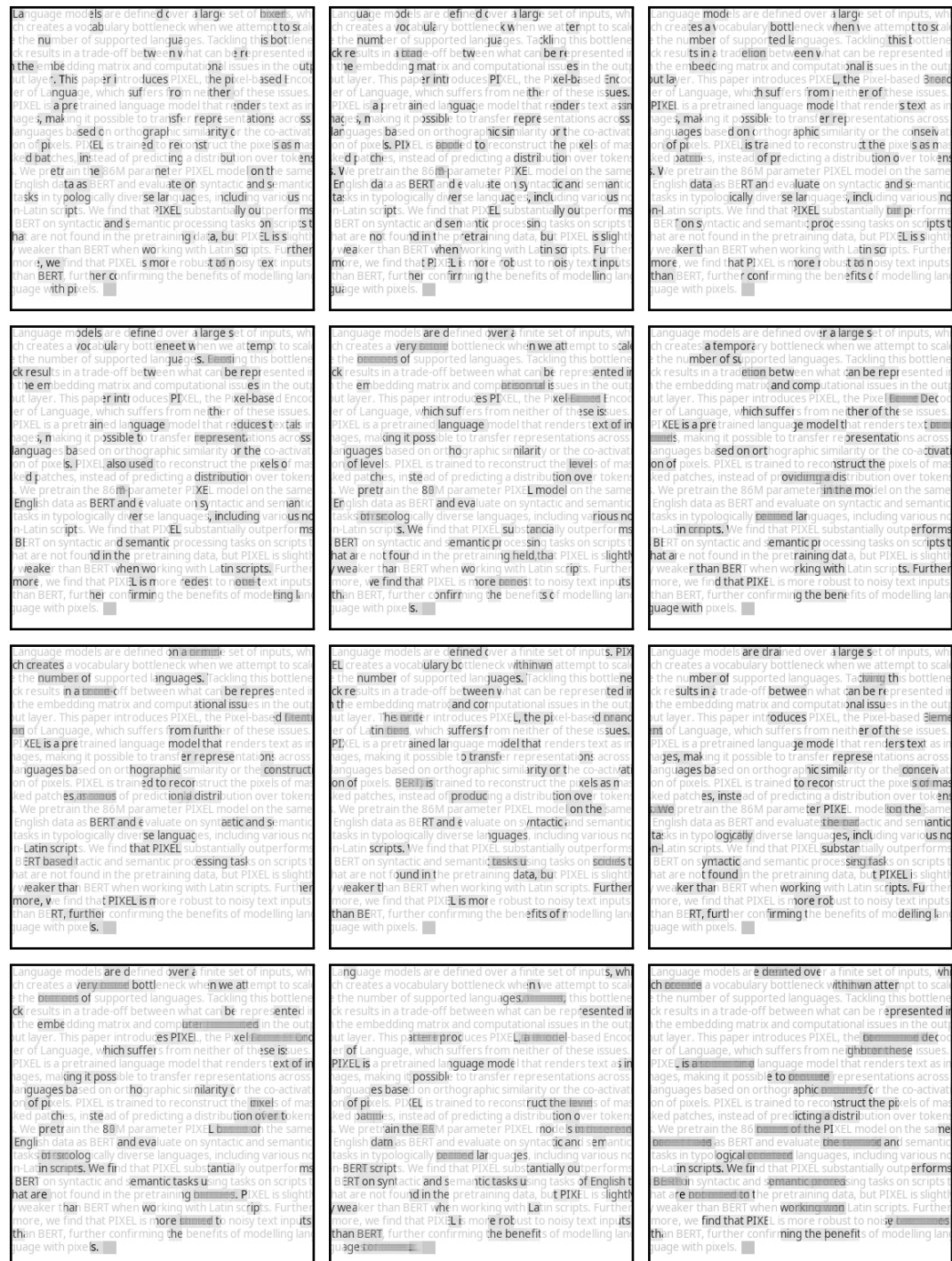

Figure 4: PIXEL image reconstructions of the abstract with different span masks.

# B  WEB TEXT RECONSTRUCTIONS

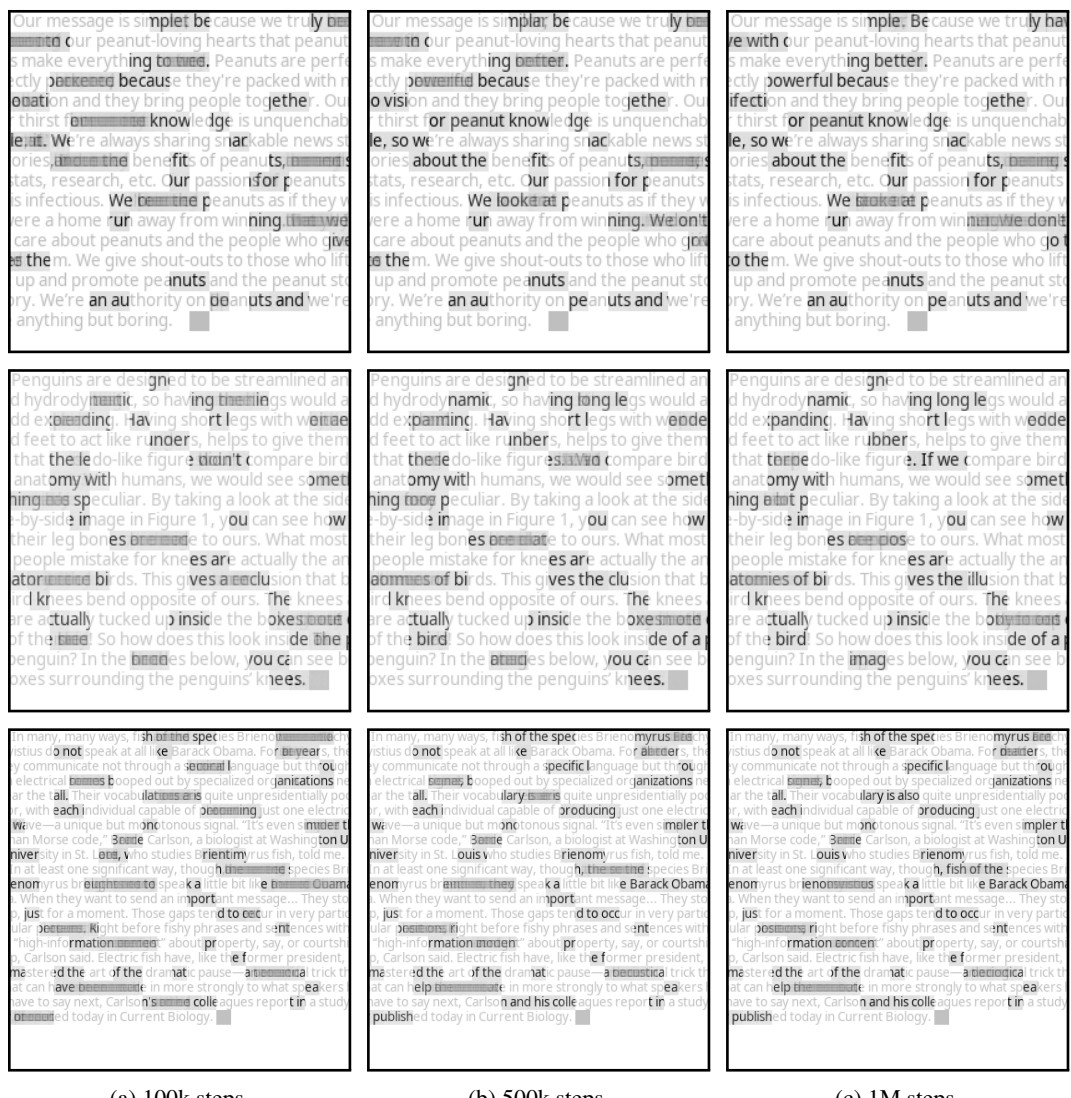

(a) 100k steps  (b) 500k steps  (c) 1M steps

Figure 5: PIXEL image reconstructions after 100k, 500k, and 1M steps of pretraining. We overlay the masked original image with the model's predictions. Images are wrapped into squares and resized for visualization purposes only. The texts were not part of the training data. We see that the fully trained PIXEL (1M) predicts masked spans more clearly and accurately. For longer spans with a larger possible prediction space, multiple predictions may appear together creating blurred text.

Reconstructions of three sources of text[19] [20] [21] after 100K, 500K and 1M pretraining steps. The figure also shows how PIXEL (visually) expresses uncertainty, e.g. for reconstructions of long spans where the space of possible outputs is much larger than for short spans, and how it captures long-range dependencies. In the third row, we can for instance see that PIXEL uses context from the beginning of a sequence (*Barack Obama*) to correctly fill in a gap later in the sequence, and vice-versa (*Brienomyrus*).

---

[19]https://www.nationalpeanutboard.org/peanut-info/our-message.htm

[20]https://www.penguinsinternational.org/2019/07/10/do-penguins-have-knees-and-other-frequently-asked-questions/

[21]https://www.theatlantic.com/science/archive/2021/05/electric-fish-pause/618993/

## C  CODE

PIXEL is implemented in PyTorch (Paszke et al., 2019) and built on HuggingFace transformers (Wolf et al., 2020). We make our code available at `https://github.com/xplip/pixel`. Our pretrained PIXEL model, including a large number of intermediate checkpoints, is available at `https://huggingface.co/Team-PIXEL/pixel-base` and our finetuned models, including multiple seeds each, are available through the model hub.

## D  TEXT RENDERER DETAILS

**Rendering backend**   We experimented with different text rendering backends. Following Salesky et al. (2021), our first implementation was based on PyGame,[22] which PIXEL was also pretrained with. Later on, we switched to a backend based on Pango (Taylor, 2004) and Cairographics,[23] which has native support for complex text layouts, making it possible to specify fallback fonts, and has faster rendering speed. Without fallback fonts, we would be limited to a maximum number of $2^{16}-1$ glyphs that can fit into a single OpenType or TrueType font file due to a technical limitation.[24] By leveraging fallback fonts, we can theoretically cover all Unicode codepoints, including emojis.

**Fonts**   We rely on the Google Noto Sans fonts collection,[25] which covers the majority of Unicode codepoints and is actively growing.[26]. Note, however, that PIXEL is compatible with any font and can therefore encode anything that can be typeset on a computer screen. We used a font size of 8 at 120 DPI for pretraining with PyGame, which was selected manually to fit most scripts into a rendered height of 16px. It can, however, also be adjusted at finetuning time. For finetuning with PangoCairo, we use a font size of $8 \cdot (120/72) \approx 13.33$ which yields roughly the same outputs as the PyGame renderer. Due to how glyphs are shaped by the two backends, the outputs of the two renderers do not *exactly* match. Because we did not employ data augmentation to make PIXEL robust to such changes in font size, we recommend using the PyGame renderer it was pretrained with for *zero-shot* applications with PIXEL. When finetuning, this minor mismatch in rendering outputs is easily overcome by PIXEL, so we generally recommend using the PangoCairo renderer.

**Characters versus glyphs**   For extractive QA, it is necessary to obtain a mapping between the characters in the context paragraph and where they appear on the rendered image. Obtaining this mapping is not straightforward due to how text is rendered. The *shaping* step in the rendering pipeline converts characters into glyphs.[27] In ligatures, as common for instance in Arabic, a glyph is composed of multiple characters. Likewise, an emoji often consists of a base codepoint and a modifier codepoint (e.g. to change the emoji skin colour) which are represented by a single glyph. For accents, on the other hand, one character might yield multiple glyphs.[28] In practice, the renderer therefore uses grapheme clusters, whose logical boundaries in the rendered image we can map to the input characters.[29] For simplicity, we assign each codepoint of a grapheme cluster to the logical horizontal offset at which the cluster starts on the rendered image. Future work may investigate alternative mapping strategies.

**RGB rendering**   PIXEL supports RGB rendering which may be useful to accurately represent colour emoji and for multimodal applications in the future. However, 24-bit RGB rendering is slightly slower than 8-bit grayscale rendering (see Table 6 below) for text written in Latin script, which is why we made RGB rendering an optional setting. In our pretraining and finetuning experiments we rendered text in grayscale, and we generally recommend doing so when not working with coloured inputs.

---

[22]`https://www.pygame.org/`

[23]`https://www.cairographics.org/`

[24]See `https://en.wikipedia.org/wiki/Unicode_font` for an explanation.

[25]`https://fonts.google.com/noto`

[26]See `https://notofonts.github.io/overview/` for an overview of Noto's Unicode coverage.

[27]See `https://docs.gtk.org/Pango/pango_rendering.html` for an overview of the rendering pipeline.

[28]`https://docs.gtk.org/Pango/pango_fonts.html#glyphs`

[29]`https://unicode.org/reports/tr29/#Grapheme_Cluster_Boundaries`

**Right-to-left scripts**    PIXEL's renderer natively supports right-to-left (RTL) writing. In the default setting, the base text direction (which for instance determines on which side of a sentence punctuation marks are placed) is inferred automatically by the rendering backend based on the first "strong directional" character in a given paragraph.[30] The mirroring of RTL characters is also handled automatically according to their Unicode bidi attributes. Optionally, the base text direction can be set manually, which is useful when working on monolingual data, e.g. in Arabic or Hebrew, as the renderer does not have to go through the direction check. In §J, we describe limitations of how we currently handle RTL writing.

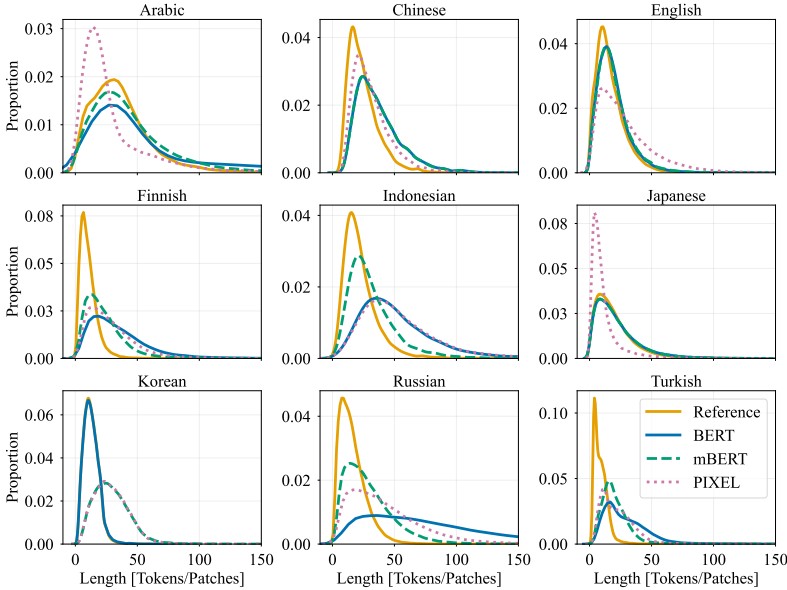

Figure 6: Distributions of sentence lengths from monolingual UD corpora after tokenizing by BERT and MBERT and rendering by PIXEL, compared to the reference by UD treebank annotators.

| Processor | Batched | Throughput [ex / s] | |
| --- | --- | --- | --- |
| | | ENG | ZHO |
| Renderer (Grayscale) | ✗ | 3944.1 | 6309.0 |
| Renderer (RGB) | ✗ | 3615.1 | 6849.5 |
| Tokenizer (Rust) | ✓ | **19128.9** | **18550.5** |
| | ✗ | 4782.9 | 5684.4 |
| Tokenizer (Python) | ✓ | 1286.6 | 2637.1 |
| | ✗ | 1286.8 | 2580.9 |

Table 6: Throughput comparison between PIXEL's PangoCairo renderer and the fast and slow BERT tokenizers, implemented in Rust and Python respectively, from the HuggingFace tokenizers library. We estimate throughput, measured in examples per second, by how long it takes to process 1M lines of English (ENG) and Chinese (ZHO) Wikipedia text on the same desktop workstation (AMD Ryzen 9 3900X 12-core CPU). We distinguish between tokenizing all lines individually (Batched = ✗) and as one single batch (✓).

**Efficiency analysis**    We briefly analyze the text processing (rendering versus tokenizing) efficiency in terms of a) length of the processed sequence, which has a direct effect on GPU memory consumption and the time it takes to compute forward and backward passes, and b) processing throughput.

For a), we follow Rust et al. (2021) and process the training and validation splits of all available UD v2.10 treebanks in various languages with the PIXEL renderer and the tokenizers of BERT and MBERT. We plot the resulting sentence length distributions in Figure 6, including a comparison

---

[30]See https://unicode.org/reports/tr9/ for an overview of the Unicode bidi algorithm.

with the reference segmentations from the UD annotators. For English text, the PIXEL renderer is slightly less efficient, i.e., it produces slightly longer sequences on average than the tokenizers. For other languages with Latin script, e.g. Finnish and Turkish, the renderer is more efficient than the BERT tokenizer, albeit slightly less efficient than the MBERT tokenizer. For non-Latin scripts such as Arabic and Japanese, we see that the renderer can be a lot more efficient than both tokenizers. The English BERT tokenizer is technically fairly space-efficient for non-Latin scripts but this is misleading because it largely produces [UNK]s (recall right side of Table 1) and each [UNK] is a single token; the functionality of the BERT model on a sequence of [UNK] is strongly compromised.

For b), we compare the processing throughput of HuggingFace's BERT tokenizers and our PIXEL renderer in Table 6. We find that the Rust-based BERT tokenizer with batch processing achieves the highest throughput by leveraging parallelization. When not using batch processing, it is comparable in throughput with PIXEL's renderer, i.e. depending on the language or script, rendering can be slightly slower (ENG) or faster (ZHO) than tokenizing. Since the rendering backend (PangoCairo) is implemented in C, we expect to achieve similar gains in rendering throughput by also leveraging parallelization for batch processing (in contrast to the Python-based tokenizer which is limited by Python's global interpreter lock (GIL)). We plan to implement batch rendering functionality in the future.

# E  ARCHITECTURE & PRETRAINING DETAILS

| PARAMETER | VALUE |
|---|---|
| Image size | (16, 8464, 3) |
| Patch size $P$ | 16 |
| Encoder hidden size $D_{\text{enc}}$ | 768 |
| Encoder intermediate size | 3072 |
| Encoder num attention heads | 12 |
| Encoder num layers $L$ | 12 |
| Decoder hidden size $D_{\text{dec}}$ | 512 |
| Decoder intermediate size | 2048 |
| Decoder num attention heads | 16 |
| Decoder num layers $K$ | 8 |
| Layer norm $\varepsilon$ (Ba et al., 2016) | $1e{-}12$ |
| Span masking ratio $R$ | 0.25 |
| Span masking max length $S$ | 6 |
| Span masking cumulative weights $W$ | $\{0.2, 0.4, 0.6, 0.8, 0.9, 1\}$ |
| Span masking spacing | Dynamic |
| Dropout probability | 0.1 |
| Hidden activation | GeLU (Hendrycks & Gimpel, 2016) |
| Optimizer | AdamW (Loshchilov & Hutter, 2019; Kingma & Ba, 2015) |
| Adam $\beta$ | (0.9, 0.999) |
| Adam $\varepsilon$ | $1e{-}8$ |
| Weight decay | 0.05 |
| Peak learning rate | $1.5e{-}4$ |
| Learning rate schedule | Cosine Decay (Loshchilov & Hutter, 2017) |
| Minimum learning rate | $1e{-}5$ |
| Learning rate warmup ratio | 0.05 |
| Training steps | 1M |
| Batch size | 256 |

Table 7: PIXEL pretraining settings

**Patch Embeddings**  PIXEL reshapes each image $x$ into a sequence of $N = W/P$ non-overlapping flattened 2D patches $x_f \in \mathbb{R}^{N \times (P^2 C)}$, where $P = 16$ is the patch size, and linearly projects them via $E \in \mathbb{R}^{(P^2 C) \times D_{\text{enc}}}$ to obtain patch embeddings $x_p = (x_f E) \in \mathbb{R}^{N \times D_{\text{enc}}}$ with encoder hidden size $D_{\text{enc}} = P^2 C = 768$.[31] Afterwards, fixed sinusoidal position embeddings $E_{\text{pos}} \in \mathbb{R}^{(N+1) \times D_{\text{enc}}}$ are added, leaving out the position vector in position 0 for a classification (CLS) embedding later:
$$\tilde{x}_p = x_p + [E_{\text{pos}}^1, \ldots, E_{\text{pos}}^{(N+1)}].$$

---

[31]This is equivalent to projecting each rendered image $x \in \mathbb{R}^{H \times W \times C}$ via a 2D-convolutional layer with $C$ input channels and $D_{\text{enc}}$ output channels and kernel size and stride both equal to the patch size $P$, which we do in practice.

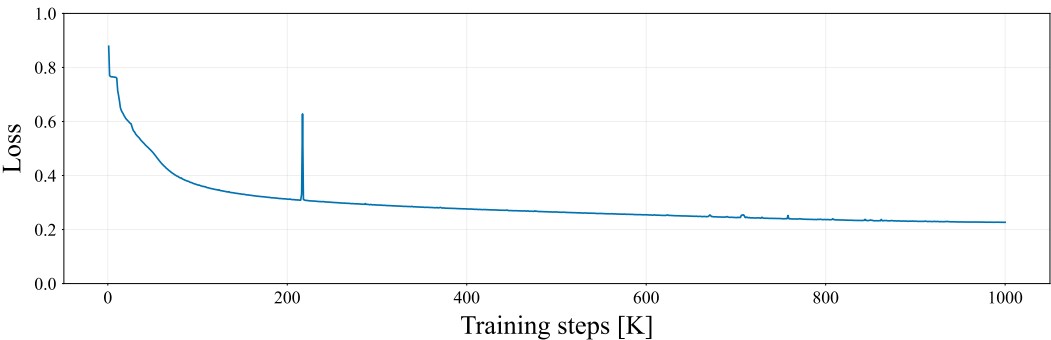

Figure 7: PIXEL pretraining loss curve

**Span Masking**   PIXEL then masks out $R = 25\%$ of the $N = 529$ embedded patches via span masking with max span length $S = 6$ and cumulative span weights $W = \{0.2, 0.4, 0.6, 0.8, 0.9, 1\}$, i.e. $\mathbb{E}(s) = 3.1$, as outlined in Algorithm 1. Applying the mask $\mathcal{M}$, we obtain the unmasked patches $\tilde{\boldsymbol{x}}_{\text{vis}} = \{\tilde{\boldsymbol{x}}_p^i : i \notin \mathcal{M}\}_{i=0}^N$.

**Encoder**   Following ViT-MAE (He et al., 2022), the PIXEL encoder only operates on unmasked patches (i.e., $\approx 396$ patches at 25% masking) and a special CLS embedding with its positional encoding $\boldsymbol{c} = \boldsymbol{x}_{[\text{cls}]} + \boldsymbol{E}_{\text{pos}}^0 \in \mathbb{R}^{1 \times D_{\text{enc}}}$ is prepended to the sequence: $\boldsymbol{h}_0 = [\boldsymbol{c}, \tilde{\boldsymbol{x}}_{\text{vis}}] \in \mathbb{R}^{(1+\lfloor R \cdot N \rfloor) \times D_{\text{enc}}}$.[32] Let $\{\boldsymbol{h}_i\}_{i=1}^L$ be the encoder hidden states after each of the $L = 12$ encoder transformer layers, and $\boldsymbol{h}_0$ denotes the input sequence. The outputs of each transformer layer are computed as detailed in (Vaswani et al., 2017),[33] and the last layer's output $\boldsymbol{h}_L \in \mathbb{R}^{(1+\lfloor R \cdot N \rfloor) \times D_{\text{enc}}}$ is passed to the decoder.

**Decoder**   The PIXEL decoder first projects the encoder outputs via $\boldsymbol{E}_{\text{dec}} \in \mathbb{R}^{D_{\text{enc}} \times D_{\text{dec}}}$ to obtain decoder embeddings $\boldsymbol{x}_d = \boldsymbol{h}_L \boldsymbol{E}_{\text{dec}} \in \mathbb{R}^{(1+\lfloor R \cdot N \rfloor) \times D_{\text{dec}}}$, where $D_{\text{dec}} = 512$. Next, mask embeddings $x_{[\text{mask}]} \in \mathbb{R}^{1 \times D_{\text{dec}}}$ are inserted at the masked-out positions and fixed sinusoidal position embeddings are added to obtain $\boldsymbol{d}_0 = [(\boldsymbol{x}_d \cup \{x_{[\text{mask}]} : i \in \mathcal{M}\}_{i=0}^N) + \boldsymbol{E}_{\text{pos}}] \in \mathbb{R}^{(N+1) \times D_{\text{dec}}}$. $\{\boldsymbol{d}_i\}_{i=1}^K$ are the decoder hidden states after each of the $K = 8$ decoder transformer layers, computed in the same way as the encoder hidden states, and $\boldsymbol{d}_0$ denotes the input sequence. There is no encoder-decoder cross-attention. The decoder output $\boldsymbol{d}_K \in \mathbb{R}^{(N+1) \times D_{\text{dec}}}$ is projected via $\boldsymbol{O} \in \mathbb{R}^{D_{\text{dec}} \times (P^2 C)}$ to obtain patch-wise logits $\boldsymbol{o} = (\boldsymbol{d}_K \boldsymbol{O}) \in \mathbb{R}^{(N+1) \times (P^2 C)}$. Finally, the CLS logits are removed and a normalized mean squared error (MSE) pixel reconstruction loss is computed: $\mathcal{L}_{\text{normpix}} = \frac{1}{|Q|} \sum_{i \in Q} |\text{normalize}(\boldsymbol{x}_f^i) - \boldsymbol{o}^i|^2$ with $i$ denoting the indices in the set of *masked, non-blank (text)* patches $Q = \{i : i \in (\mathcal{M} \cap \mathcal{T})\}_{i=0}^N$ and normalize$(\cdot)$ dividing the difference between the target patch and its mean by its standard deviation.

## F  FINETUNING DETAILS

Table 8 gives an overview of all languages used in our finetuning experiments, Table 9 links to our finetuning datasets, and Table 10 lists the UD treebanks we used.

We list our finetuning recipes in Table 11 for POS tagging, dependency parsing, NER, QA, and XNLI and in Table 12 for the GLUE tasks. Due to compute limitations we did not run comprehensive hyperparameter sweeps. Instead, we relied on sensible priors from finetuning BERT and made slight modifications as needed. In most cases, hyperparameters that work well for BERT also work well for PIXEL. For some of the semantic tasks, in particular NLI and SST-2, we found that some random initializations did not converge. In those cases, minor tweaks to the learning rate or increasing the batch size usually helped. For GLUE, we found that PIXEL performed slightly better on some tasks with the PangoCairo renderer, whereas for others, using the PyGame renderer (which PIXEL was

---

[32]In pretraining, no loss is computed for the CLS embedding but it can optionally be used when finetuning PIXEL for sequence-level downstream tasks.

[33]Note that encoder and decoder do not attend to the blank (padding) patches that appear after the EOS patch.

pretrained with) was more stable. We plan to further optimize the training recipes and study PIXEL's convergence behaviour in the future.

For word-level tasks, we add padding in order to render each word at the start of a new image patch and so create a bijective mapping between words and patches. Doing so assumes that word boundaries are available. We note that subword-based and character-based models also make this assumption. In BERT, for instance, word-level tasks are formulated such that a word's label is assigned to its first subword token, requiring word boundaries. During training, continuation tokens are then masked out when computing the loss. Consequently, predictions for continuation tokens also need to be masked out at inference time, which again requires word boundaries or aggregation strategies that may introduce errors. The same applies to character-based models. For PIXEL, should this assumption be violated, it is still possible to render the text without adding spacing, although the mapping is then no longer bijective as multiple words can overlap on one image patch. In such cases, assigning the prediction for a patch to either word can cause loss of information. Although in practice this approach does not necessarily affect performance negatively, future work will investigate alternative approaches.

| Language | ISO 639-3 | Language Family | Script |
|---|---|---|---|
| Amharic | AMH | Afro-Asiatic | Ge'ez |
| Arabic | ARA | Afro-Asiatic | Arabic |
| Bengali | BEN | Indo-European | Bengali |
| Bulgarian | BUL | Indo-European | Cyrillic |
| Chinese | ZHO | Sino-Tibetan | Chinese |
| Coptic | COP | Afro-Asiatic | Coptic |
| English | ENG | Indo-European | Latin |
| Finnish | FIN | Uralic | Latin |
| French | FRA | Indo-European | Latin |
| German | DEU | Indo-European | Latin |
| Greek | ELL | Indo-European | Greek |
| Hausa | HAU | Afro-Asiatic | Latin |
| Hindi | HIN | Indo-European | Devanagari |
| Igbo | IBO | Niger-Congo | Latin |
| Indonesian | IND | Austronesian | Latin |
| Japanese | JPN | Japonic | Japanese |
| Kinyarwanda | KIN | Niger-Congo | Latin |
| Korean | KOR | Koreanic | Korean |
| Luganda | LUG | Niger-Congo | Latin |
| Luo | LUO | Nilo-Saharan | Latin |
| Naija Pidgin | PCM | English Creole | Latin |
| Russian | RUS | Indo-European | Cyrillic |
| Spanish | SPA | Indo-European | Latin |
| Swahili | SWA | Niger-Congo | Latin |
| Tamil | TAM | Dravidian | Tamil |
| Telugu | TEL | Dravidian | Telugu |
| Thai | THA | Kra-Dai | Thai |
| Turkish | TUR | Turkic | Latin |
| Urdu | URD | Indo-European | Perso-Arabic |
| Vietnamese | VIE | Austro-Asiatic | Latin |
| Wolof | WOL | Niger-Congo | Latin |
| Yorùbá | YOR | Niger-Congo | Latin |

Table 8: Overview of languages used in our experiments.

| Dataset | Download Link | Reference |
|---|---|---|
| Universal Dependencies 2.10 | https://lindat.mff.cuni.cz/repository/xmlui/handle/11234/1-4758 | (Zeman et al., 2022; Nivre et al., 2020) |
| MasakhaNER | https://github.com/masakhane-io/masakhane-ner/tree/main/data | (Adelani et al., 2021) |
| GLUE | https://huggingface.co/datasets/glue | (Wang et al., 2018) |
| TyDiQA-GoldP | https://huggingface.co/datasets/tydiqa | (Clark et al., 2020) |
| SQuADv1.1 | https://huggingface.co/datasets/squad | (Rajpurkar et al., 2016) |
| KorQuAD 1.0 | https://huggingface.co/datasets/squad_kor_v1 | (Lim et al., 2019) |
| JaQuAD | https://huggingface.co/datasets/SkelterLabsInc/JaQuAD | (So et al., 2022) |
| XNLI | https://huggingface.co/datasets/xnli | (Conneau et al., 2018) |

Table 9: Links and references to the datasets we used in our finetuning experiments.

| Language | Treebank | #Sentences | Reference |
|---|---|---|---|
| ENG | English-EWT | 16621 | Silveira et al. (2014) |
| ARA | Arabic-PADT | 7664 | Hajič et al. (2009) |
| COP | Coptic-Scriptorium | 2011 | Zeldes & Abrams (2018) |
| HIN | Hindi-HDTB | 16647 | Palmer et al. (2009) |
| JPN | Japanese-GSD | 8100 | Asahara et al. (2018) |
| KOR | Korean-GSD | 6339 | Chun et al. (2018) |
| TAM | Tamil-TTB | 600 | Ramasamy & Žabokrtský (2012) |
| VIE | Vietnamese-VTB | 3000 | Nguyen et al. (2009) |
| ZHO | Chinese-GSD | 4997 | Shen et al. (2016) |

Table 10: Overview of the Universal Dependencies v2.10 (Zeman et al., 2022; Nivre et al., 2020) treebanks used in our POS tagging and dependency parsing experiments with the number of sentences in their respective training splits. As mentioned in §3.1, these treebanks were chosen with typological and script diversity in mind.

| PARAMETER | POS | DP | NER | QA | XNLI |
|---|---|---|---|---|---|
| Rendering backend | | | PangoCairo | | |
| Classification head pooling | — | — | — | — | CLS |
| Optimizer | | | AdamW | | |
| Adam $\beta$ | | | (0.9, 0.999) | | |
| Adam $\varepsilon$ | | | $1e-8$ | | |
| Weight decay | | | 0 | | |
| Learning rate | $5e-5$ | $\{5e-5, 8e-5\}$ | $5e-5$ | $\{3e-5, 5e-5, 7e-5\}$ | $2e-5$ |
| Learning rate warmup steps | 100 | 100 | 100 | 100 | 1000 |
| Learning rate schedule | | | Linear decay | | |
| Max sequence length | 256 | 256 | 196 | 400 | 196 |
| Stride | — | — | — | 160 | — |
| Batch size | 64 | 64 | 64 | 32 | 256 |
| Max steps | 15000 | 15000 | 15000 | 20000 | 50000 |
| Early stopping | | | ✓ | | |
| Eval steps | 500 | 500 | 500 | 500 | 1000 |
| Dropout probability | | | 0.1 | | |

Table 11: Finetuning settings for POS tagging, dependency parsing (DP), NER, QA, and XNLI. We did not run a comprehensive hyperparameter search due to compute limitations; these settings were manually selected based on a small number of preliminary runs. Maximum performance was often reached well before the specified number of max steps.

| PARAMETER | MNLI | QQP | QNLI | SST-2 | COLA | STS-B | MRPC | RTE | WNLI |
|---|---|---|---|---|---|---|---|---|---|
| Rendering backend | PangoCairo | PyGame | PangoCairo | PyGame | PyGame | PyGame | PyGame | PyGame | PyGame |
| Classification head pooling | | | | | Mean | | | | |
| Optimizer | | | | | AdamW | | | | |
| Adam $\beta$ | | | | | (0.9, 0.999) | | | | |
| Adam $\varepsilon$ | | | | | $1e-8$ | | | | |
| Weight decay | | | | | 0 | | | | |
| Learning rate | $3e-5$ | $3e-5$ | $3e-5$ | $3e-5$ | $2e-5$ | $2e-5$ | $3e-5$ | $3e-5$ | $1e-5$ |
| Learning rate warmup steps | 100 | 100 | 100 | 100 | 200 | 100 | 100 | 200 | 100 |
| Learning rate schedule | | | | | Linear decay | | | | |
| Max sequence length | | | | | 256 | | | | |
| Batch size | 64 | 256 | 64 | 256 | 256 | 64 | 64 | 64 | 256 |
| Max steps | 15000 | 15000 | 15000 | 15000 | 15000 | 15000 | 15000 | 15000 | 400 |
| Early stopping | | | | | ✓ | | | | |
| Eval interval | 500 steps | 500 steps | 500 steps | 500 steps | 100 steps | 100 steps | 100 steps | 250 steps | 1 epoch |
| Dropout probability | | | | | 0.1 | | | | |

Table 12: Finetuning settings for GLUE tasks. We did not run a comprehensive hyperparameter search due to compute limitations; these settings were manually selected based on a small number of preliminary runs. Increasing the batch size to 256 and switching to the PyGame renderer helped achieve more consistent convergence behaviour for some tasks. For the smaller datasets (to the right of QQP), maximum performance was reached well before the specified number of max steps.

## G    EXAMPLES OF *Zeroé* ORTHOGRAPHIC ATTACKS

| Attack | Sentence |
|---|---|
| NONE | Penguins are designed to be streamlined |
| CONFUSABLE | Penguins are designed to be streamlined |
| SHUFFLE (INNER) | Pegnuins are dnesiged to be sieatrnmled |
| SHUFFLE (FULL) | ngePnius rae dsgednei to be etimaslernd |
| DISEMVOWEL | Pngns r dsgnd to be strmlnd |
| INTRUDE | Pe'nguins a{re d)esigned t;o b*e stre<amlined |
| KEYBOARD TYPO | Penguinz xre dwsigned ro ne streamllned |
| NATURAL NOISE | Penguijs ard design4d ti bd streamlinfd |
| TRUNCATE | Penguin are designe to be streamline |
| SEGMENTATION | Penguinsaredesignedtobestreamlined |
| PHONETIC | Pengwains's ar dhiseind te be storimlignd |

Table 13: Examples of low-level orthographic attacks based on the *Zeroé* benchmark.

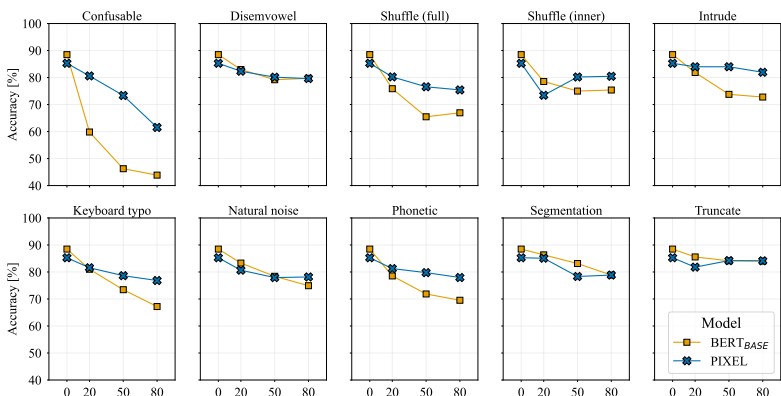

Figure 8: Test set accuracy for a single run of PIXEL and BERT across different levels of noise introduced through various orthographic attacks in SNLI. The results show that PIXEL is more robust than BERT to most of these attacks.

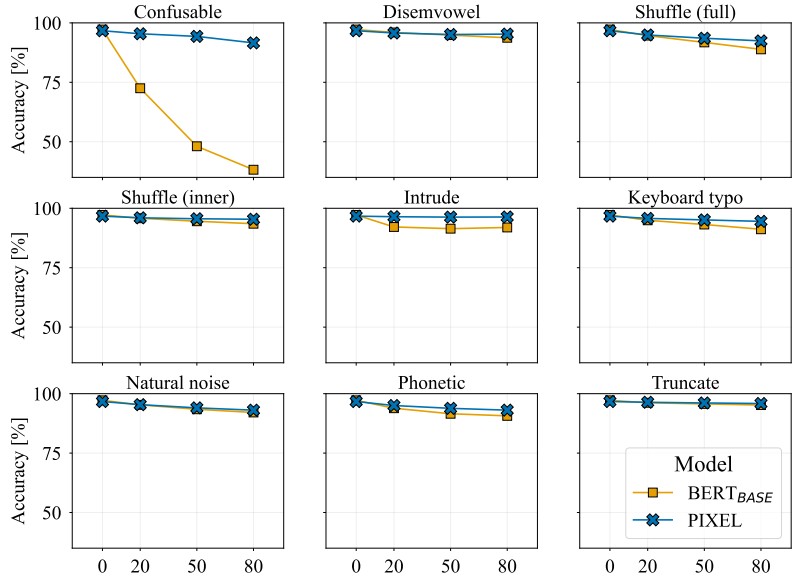

Figure 9: Test set accuracy for a single run of PIXEL and BERT across different levels of noise introduced through various orthographic attacks in POS tagging. The results show that PIXEL is more robust than BERT to most of these attacks, especially when dealing with visually-confusable character substitutions. SEGMENTATION is not applied to the task of POS tagging, since the joined words would not have a proper tag.

## H  FONT TRANSFER ANALYSIS

In this section, we analyse the adaptation capabilities of PIXEL to new fonts at finetuning time. Specifically, we finetune PIXEL models for POS tagging and dependency parsing on the UD_English-EWT treebank and sentiment analysis on SST-2, once with a font similar to our `GoNotoCurrent` / `NotoSans-Regular` pretraining font, `NotoSerif-Regular`, and once with a font strikingly different from it, `JournalDingbats1`. We compare the three fonts in Table 14 below:

| Font | Rendered Example Sentence |
|------|---------------------------|
| GoNotoCurrent | My cat loves oatmeal and pancakes. ■ |
| NotoSerif-Regular | My cat loves oatmeal and pancakes. ■ |
| JournalDingbats1 | ⬡⬢✶⬟⬢✹⬡⬢✶⬟⬡⬢✹⬡⬢▲⬡⬢⬟▲⬡⬢⬟⬡⬢✹⬡⬢▲⬡■ |

Table 14: An example sentence rendered in three different fonts.

|       | GoNotoCurrent | NotoSerif-Regular | JournalDingbats1 |
|-------|---------------|-------------------|------------------|
| POS   | 96.7          | 95.9              | 93.9             |
| DP    | 90.6          | 88.1              | 81.3             |
| SST-2 | 89.6          | 84.2              | 72.9             |

Table 15: Results for fine-tuning PIXEL for POS tagging, dependency parsing (DP), and sentiment analysis on SST-2 with three different fonts: the font used in pretraining (`GoNotoCurrent`), a visually similar font (`NotoSerif-Regular`), and a highly dissimilar font (`JournalDingbats1`). We report test accuracy for POS, test LAS for DP, and validation accuracy for SST-2, each averaged over 5 runs.

The font transfer results are shown in Table 15. We find that PIXEL exhibits fairly high font transfer ability *out-of-the-box*, i.e. without any font or image augmentation strategies employed during pretraining.[34] In line with our expectations, transfer to a visually similar font (`NotoSerif-Regular`) is easier than to a dissimilar font (`JournalDingbats1`). Nevertheless, PIXEL is able to transfer surprisingly well to the `JournalDingbats1` font, in which every letter is simply mapped to the icon of an object or animal.

## I  FURTHER ANALYSIS

To investigate where PIXEL currently lags behind BERT, we analyse the impact that dependency length has on both models in dependency parsing in ENG. We can see in Figure 10 that the LAS gap between BERT and PIXEL increases with longer dependencies, indicating that PIXEL struggles slightly more with long syntactic dependencies.

|       | #L  | θ    | ENG  | ARA  | BUL  | DEU  | ELL  | FRA  | HIN  | RUS  | SPA  | SWA  | THA  | TUR  | URD  | VIE  | ZHO  |
|-------|-----|------|------|------|------|------|------|------|------|------|------|------|------|------|------|------|------|
| MBERT | 104 | 179M | 83.3 | 73.2 | 77.9 | 78.1 | 75.8 | 78.5 | 70.1 | 76.5 | 79.7 | 67.2 | 67.7 | 73.3 | 66.1 | 77.2 | 77.7 |
| BERT  | 1   | 110M | **83.7** | **64.8** | **69.1** | **70.4** | **67.7** | **72.4** | **59.2** | **66.4** | **72.4** | **62.2** | 35.7 | **66.3** | **54.5** | **67.6** | 46.2 |
| PIXEL | 1   | 86M  | 77.2 | 58.9 | 66.5 | 68.0 | 64.9 | 69.4 | 57.8 | 63.4 | 70.3 | 60.8 | **50.2** | 64.0 | 54.1 | 64.8 | **52.0** |

Table 16: Results for PIXEL and BERT finetuned on XNLI in the *translate-train-all* setting where we train on the joint training data in all 15 languages, originally translated from ENG by Conneau et al. (2018). We report test set accuracy averaged over 5 runs each. Despite the relatively large performance gap in favor of BERT in ENG (which is in line with the GLUE results in Table 3), the gap is much smaller for other languages, particularly those not using the Latin writing system. PIXEL is overall more consistent across scripts, outperforming BERT in THA and ZHO.

---

[34]We believe such augmentation strategies would further improve robustness to font variations and leave this experiment to future work. Considering that we have full control over the font when working with NLP text datasets, robustness to font variations was not a primary goal in this work.

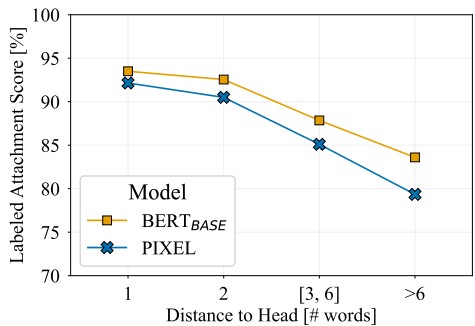

Figure 10: LAS scores (ENG) across different dependency lengths averaged over 5 random intitial-izations of BERT and PIXEL. In ENG, long syntactic dependencies are more challenging for PIXEL.

## J   LIMITATIONS

This paper introduces a new approach to processing written language as images, which removes the need for a finite vocabulary, providing a solution to the *vocabulary bottleneck*. While our results show that PIXEL is a promising approach in this direction, this is only the first step. Here, we highlight current limitations and avenues for future work for pixel-based models:

- PIXEL is pretrained on predominantly English text written in the Latin script. The choice of English is driven by the scientific goal of comparing against a widely used model (English BERT) but English may not be the best source language for cross-lingual transfer (Turc et al., 2021; Blevins et al., 2022). We expect that PIXEL trained on typologically diverse languages in multiple scripts would considerably surpass the cross-script and cross-lingual transferability of English-only PIXEL but this remains to be verified, and training a model on large amounts of data will require large computational resources.
- PIXEL currently seems to be less sample-efficient than subword-based PLMs. PIXEL excels at syntactic tasks after being pretrained for the same number of steps/datapoints as BERT (a challenging setup within an academic budget), but still lags behind in semantic processing. As a consequence, it also requires more training steps than BERT to converge during finetuning. Closing this gap might involve longer pretraining with additional (long-dependency) objectives.
- There are challenges to be addressed when working with languages written right-to-left. PIXEL currently processes sentences in such languages from the end to the beginning which may lead to learning inadequate features for sentence separation and position embeddings.
- PIXEL cannot be used for language generation tasks because it is not possible to produce discrete words from the pretrained decoder.
- Rendering text as images requires more disk space than reading text from a file. This can be alleviated by caching the dataset in a compressed format, or rendering the images on-the-fly. Rendering images on-the-fly will create additional overhead when training for multiple epochs.

