# OpenReview forum: "Language Modelling with Pixels"
_ICLR.cc/2023/Conference — ICLR 2023 notable top 5%_

### Official Review · Reviewer_2dJC · 2022-10-21

**Confidence:** 4
**Correctness:** 3
**Technical Novelty And Significance:** 4
**Empirical Novelty And Significance:** 4
**Recommendation:** 8

**Clarity, Quality, Novelty And Reproducibility:**

The paper is generally well-written.

Questions
- For word level tasks in Latin, how to you think that a word might be splitted into two patches?
- It would be interesting if the authors could provide some examples(cases) or any other possible ways to better illustrate how it avoids the vocabulary bottleneck problem (especially in non-Latin languages).
- The authors claimed “PIXEL is more robust than BERT to orthographic attacks and linguistic code-switching”. However, “in lexical code-switching experiments, PIXEL performs on-par with BERT”. Also, in Table 5, it seems we cannot conclude that “PIXEL is more robust than BERT to linguistic code-switching”.

**Strength And Weaknesses:**

## Strength
- Avoid vocabulary bottleneck problem.
- Show more robustness to orthographic attacks and code-switching
- Perform better on non-Latin languages on syntactic tasks and semantic tasks comparing to Bert.
- Demonstrate the possibility of pixel-based language model without injecting a priori glyph knowledge.

Weaknesses
- Average performance on downstream tasks still lags far behind Bert and mBert.
- Data preprocessing (text rendering) is tricky and laborious and will cause inevitable latency in real application.
- The generated text pixels during pretraining is blurred and fuzzy which makes the model learning less explainable.

**Summary Of The Paper:**

## Overview
In this paper, PIXEL, a pixel-based encoder of language is introduced to circumvent the vocabulary bottleneck problem. PIXEL is built on ViT-MAE and is pretrained on the same corpus as BERT. Specifically, the authors proposed to render text onto blank images and patchify raw pixels into image tokens and masking 25% of the tokens. Following He et al. (2022), the author then trained PIXEL by using a decoder to reconstruct the masked pixels of the input images. The authors evaluated PIXEL on syntactic and semantic tasks in typologically diverse languages and demonstrated its robustness to orthographic attacks and linguistic code-switching. Although the downstream performance is falling short comparing with BERT and mBERT, the proposed method shows its novelty and potential by providing a new view of tackling the vocabulary bottleneck problem in both mono-lingual and multi-lingual setting.



**Summary Of The Review:**

Although it might be not that useful, I like this idea and it gives a new perspective  for text representation

---

> ### Author Response · Authors · 2022-11-18
> **Reply to reviewer 2dJC**
>
> >Average performance on downstream tasks still lags far behind Bert and mBert.
>
> Yes, there is no denying that PIXEL has a lot of room for improvement in some of the downstream applications. We nevertheless hope to provide a strong proof-of-concept for pixel-based language representations and a useful starting point for further improvements.
>
> >Data preprocessing (text rendering) is tricky and laborious and will cause inevitable latency in real application.
>
> We think that writing a renderer is not substantially more laborious than an efficient tokenizer. Consider the extensive HuggingFace Tokenizers library as an example. Ultimately, however, the tokenizer and renderer have nearly identical interfaces to someone who simply wants to use it. Rendering a piece of text for PIXEL can be as simple as calling
>
> ```
> from pixel import PangoCairoTextRenderer
>
> # Load renderer (w/ basic functionality)
> text_renderer = PangoCairoTextRenderer.from_pretrained(“blind-for-review”)
>
> # Render a single string
> example = "My cat loves oatmeal."
> encoding = text_renderer(example)
> ```
>
> We also note that our renderer has similar throughput as a single-threaded tokenizer for English text (see Table 6 in the Appendix), and can even be faster for certain scripts such as Chinese. We have added new results for this to Table 6. The main limitation of the renderer in terms of efficiency is the fact that it does not (yet) support multiprocessing.
>
> We agree, however, that rendered images stored on disk is less efficient than storing text, so ideally the rendering should be done on-the-fly. This is currently possible with the PIXEL implementation.
>
> >The generated text pixels during pretraining is blurred and fuzzy which makes the model learning less explainable.
>
> Yes, not having a softmax probability distribution over a vocabulary means it can be harder to identify what predictions the model is assigning a high probability to. We hope to find solutions for this limitation in future work.
>
> >For word level tasks in Latin, how to you think that a word might be splitted into two patches?
>
> The way a word might be split into two (or more) patches depends entirely on the length of the word.
>
> >It would be interesting if the authors could provide some examples(cases) or any other possible ways to better illustrate how it avoids the vocabulary bottleneck problem (especially in non-Latin languages).
>
> The effect of the vocabulary bottleneck on the model inputs can be seen in Table 1, where the BERT tokenizer is unable to find IDs for the majority of Coptic, Korean, Tamil, and Chinese text. This results in a tokenization that is mostly UNK tokens, which affects model performance. PIXEL overcomes the vocabulary bottleneck here because the text renderer can support these alphabets, which means the model has access to the complete inputs, hence the model is fine-tuned on the full data.
>
> This example, chosen at random from the Chinese Universal dependencies dataset, shows the incorrect tokenization by the BERT model:
>
>
> **Input:** `"動作冒險遊戲（A-AVG）：是冒險遊戲的分支，它融合了動作遊戲的一些特徵。"`
>
> **Tokenization:** `"[UNK] [UNK] [UNK] [UNK] [UNK] [UNK]（ A - AVG) : [UNK] [UNK] [UNK] [UNK] [UNK] [UNK] [UNK] [UNK] ， [UNK] [UNK] [UNK] [UNK] [UNK] [UNK] [UNK] [UNK] [UNK] 一 [UNK] [UNK] [UNK] 。"`
>
> You can see that most of the characters are not tokenized correctly. We could add further examples like this to our paper if it would help illustrate the problem of the vocabulary bottleneck on the input data.
>
> >The authors claimed “PIXEL is more robust than BERT to orthographic attacks and linguistic code-switching”. However, “in lexical code-switching experiments, PIXEL performs on-par with BERT”. Also, in Table 5, it seems we cannot conclude that “PIXEL is more robust than BERT to linguistic code-switching”.
>
> We agree that the language can be clearer and more consistent on this. In Table 5, the averages across the five results are: mBERT: 77.1, BERT: 75.8, and PIXEL: 76.5. Therefore, we conclude *on average* that PIXEL is more robust to linguistic code-switching.

---

### Official Review · Reviewer_znCA · 2022-10-22

**Confidence:** 3
**Correctness:** 4
**Technical Novelty And Significance:** 3
**Empirical Novelty And Significance:** 3
**Recommendation:** 6

**Clarity, Quality, Novelty And Reproducibility:**

Clarity
Overall it is clear.

Quality and Novelty
Overall the quality is great. The experiments are thorough and detailed. However, some ablation studies will be helpful, for example
I am curious why authors choose RGB channel? This seems not very useful in general text besides emojis.
Also I am curious if the font of the text can influence the model performance?

Also what is the computation time for the proposed method (both training and inference) for the proposed method?

Reproducibility
The experiment set up is clear. Authors promise to publish the implementation and model checkpoints and weights, so I believe it would be reproducible.

**Strength And Weaknesses:**

Strength

It is interesting to see authors propose to render the text to image for language modeling.

I believe the robust to orthographic attacks and code-switching will be very useful in web security domains. For example, such attack pattern very common in phishing email as the fraudster try to bypass the classifier.

Weaknesses


**Summary Of The Paper:**

In this paper, authors proposed a new methodology by rendering text as images, making it possible to transfer representations across languages based on orthographic similarity or the co-activation of pixels. The pre-trained step is learning reconstructing the pixels of masked patches instead of predicting a distribution over tokens.

In the empirical study, authors find that PIXEL substantially outperforms BERT on syntactic and semantic processing tasks on scripts that are not found in the pretraining data, but PIXEL is slightly weaker than BERT when working with Latin scripts.

**Summary Of The Review:**

Overall I feel this paper is novel and has strong real-world application. However, my minor concern is the novelty is limited given visual text representations is already proposed by Salesky et al. (2021). Therefore I would recommend "marginally above threshold".

---

> ### Author Response · Authors · 2022-11-11
> **Question about potential missing review content**
>
> Thank you for your review. In the current Strengths and Weaknesses section, reproduced below, we can see what you wrote about the Strengths of the paper. We also see the word "Weaknesses" but there is nothing listed underneath it. In order to prepare a complete response, we would like to ask whether this was intentional or if the review is missing some content in this section?
>
> >Strength And Weaknesses:
> >
> >Strength
> >
> >It is interesting to see authors propose to render the text to image for language modeling.
> >
> >I believe the robust to orthographic attacks and code-switching will be very useful in web security domains. For example, such attack pattern very common in phishing email as the fraudster try to bypass the classifier.
> >
> >Weaknesses

---

> ### Author Response · Authors · 2022-11-18
> **Reply to reviewer znCA**
>
> >Overall the quality is great. The experiments are thorough and detailed. However, some ablation studies will be helpful, for example I am curious why authors choose RGB channel? This seems not very useful in general text besides emojis.
>
> It is true that for most applications with general text, RGB is not very useful. The main reasons why we nevertheless decided to train an RGB model are 1) to make PIXEL representations suitable for emoji data out-of-the-box and 2) to facilitate adaptation of PIXEL to applications that require RGB inputs, for instance making it possible to work with RGB scans/screenshots or text-image pairs.
>
> >Also I am curious if the font of the text can influence the model performance?
>
> We have trained early PIXEL prototypes with the Arial Unicode MS font and did not observe a difference in performance to variants trained with Google Noto Sans. We ultimately switched to Noto due to its open font licence, broader language/script support, and active maintenance.
>
> We also added a new experiment in Appendix H of our revised paper in which we finetuned PIXEL models with fonts not seen during pretraining. (We note that PIXEL was only pretrained with one font: GoNotoCurrent.) We found that PIXEL is fairly robust to even substantial font variations, although performance will be best when sticking to the rendering used in pretraining. We note that generalisation across fonts or different renderings was not a primary goal for us here since we have full control over the rendering as long as we deal with standard NLP text data. Otherwise, applying font or image augmentation strategies during pretraining would likely make the model a lot more robust to changes in rendering, but we leave this study for future work.
>
> >Also what is the computation time for the proposed method (both training and inference) for the proposed method?
>
> Forward and backward passes take roughly the same time as BERT because PIXEL and BERT use almost identical transformer stacks, and BERT has $\mathcal{O}(1)$ embedding lookup. For rendering throughput, we refer to Table 6 in the appendix. Note that a lower rendering throughput is not an issue as long as the dataloader can prepare batches faster than the model processes them, which is generally the case for PIXEL.
> As highlighted in Appendix J (Limitations), PIXEL can require more training steps to converge than BERT (our hyperparameter details are listed in Table 11 and Table 12), however, which we hope to address in the future. Inference times are unaffected by this limitation.
>
> >However, my minor concern is the novelty is limited given visual text representations is already proposed by Salesky et al. (2021).
>
> We believe that our paper makes several novel contributions. Most importantly, it shows that visual text representations work for general-purpose language model pretraining, as opposed to machine translation only. Furthermore, the trained model can transfer across many writing systems on a variety of tasks.

---

### Official Review · Reviewer_zqCz · 2022-10-25

**Confidence:** 3
**Correctness:** 3
**Technical Novelty And Significance:** 4
**Empirical Novelty And Significance:** 4
**Recommendation:** 6

**Clarity, Quality, Novelty And Reproducibility:**

The paper is generally comprehensive, but some details are hard to find (for instance, see previous section for suggestions). It might also be worth restructuring the results section so that it highlights conclusions a bit better – it’s a bit hard to parse right now.

This paper is novel and interesting, and the experiments are comprehensive. A bit more discussion on design decisions, ablations, qualitative analyses might enhance the quality further.

The paper states that code, checkpoints etc will be shared for reproducibility.


**Strength And Weaknesses:**

Strengths:
1. This paper takes on a new and interesting approach to a very well-established masked language model pre-training/fine-tuning paradigm. It is quite an interesting and refreshing setup.
2. The paper includes an extensive set of experiments, covering POS tagging, dependency parsing, extractive QA, GLUE, XNLI, NER.
3. Being able to non-trivially handle new scripts from only fine-tuning data is an impressive feature of the system.

Weaknesses:
1. It would have been really interesting to see the byte-based vocabulary baseline (something like ByT5). Since most of the experiments involve shorter sequences (right?), is length of the resulting byte-based sequences a concern? In some cases where the vocabulary clearly has very poor coverage, comparing against BERT seems like a slightly unfair comparison with BERT mostly fine-tuning on a sequence of unks.
2. The discussion on patch span masking is a bit unclear. How was the resolution of the rendered text chosen? Within the same height and sequence length, how many words tend to end up in a patch? Or do patches often contain subwords? I may have missed these details but it seems relevant for understanding how things work. Were there any experiments that varied these parameters to clarify the decisions? A figure illustrating the approach with an example might also be helpful.
3. The robustness experiments currently lack decided conclusions/takeaways. It doesn’t seem like Pixel is firmly more robust than BERT. Also, the orthographic attack graphs might need error bars?
4. What if Pixel is provided inputs that are rendered vastly differently during fine-tuning? It would fail to generalize right? Or would the approach be to run OCR and get the text, render it with the Pixel text renderer and then predict? It might be worth adding a discussion that goes over some of these points, and more generally delves into design decisions that are crucial.

Nit: maybe the paper title should be masked language modeling with pixels.


**Summary Of The Paper:**

This paper presents Pixel, a masked language modeling approach similar to BERT that first renders text as an image and then encodes the sequence of patches using a ViT. During pre-training, the setup resembles the masked autoencoder – spans of patches are masked, only unmasked patches are encoded and a decoder is used to predict the masked patches. During fine-tuning, the decoder is replaced with a classification head similar to BERT. Pixel is motivated by the need to remove the need for large vocabularies that increase model capacity via embedding layers and make prediction layers very expensive, while being able to generalize to a large set of scripts and languages.

Experiments include pre-training Pixel on the same data as BERT (Wiki + Books) and fine-tuning on a large set of tasks including pos tagging, dependency tagging, NER, GLUE/XNLI and extractive QA. Results show that while BERT often outperforms Pixel on English tasks and latin scripts, Pixel can transfer better to previously unseen scripts. They also conduct analysis on robustness to low-level orthographic attacks and code-switching.


**Summary Of The Review:**

Recommending weak accept (open to changing this after discussions) because the paper proposes an interesting approach to masked language modeling with a comprehensive set of experiments, but there is a slight lack of clarity in a few places.

---

> ### Author Response · Authors · 2022-11-18
> **Reply to reviewer zqCz (1/2)**
>
> >It would have been really interesting to see the byte-based vocabulary baseline (something like ByT5). Since most of the experiments involve shorter sequences (right?), is length of the resulting byte-based sequences a concern? In some cases where the vocabulary clearly has very poor coverage, comparing against BERT seems like a slightly unfair comparison with BERT mostly fine-tuning on a sequence of unks.
>
> We can include results of ByT5 where the result is already reported in the ByT5 paper. We note, however, that these results are not directly comparable since ByT5 is both a larger model (300M+ parameters) and was pretrainied on large amounts of multilingual text (mC4) while PIXEL was only pretrained on English text.
>
> >The discussion on patch span masking is a bit unclear. How was the resolution of the rendered text chosen? Within the same height and sequence length, how many words tend to end up in a patch? Or do patches often contain subwords? I may have missed these details but it seems relevant for understanding how things work. Were there any experiments that varied these parameters to clarify the decisions? A figure illustrating the approach with an example might also be helpful.
>
> **Patch Span Masking is unclear:** we are not sure about what was unclear about our discussion of patch span masking. What would improve this section of the paper? Would it help if we added visualisations of the output Algorithm 1?
>
> **Resolution:** The patch resolution was chosen to be 16x16 because that is the standard resolution used by most vision transformer architectures (ViT, ViT-MAE, etc.). The image dimensions were thus chosen to be 16x8464 which corresponds to 529 non-overlapping 16x16 patches. Why did we want a sequence length of 529? It is the closest sequence length to BERT’s 512 (i.e., the most comparable), while also satisfying the (somewhat arbitrary but nice-to-have) property that the square root of the sequence length is an integer value, i.e. $sqrt(529) = 23$, as this allows to optionally reshape the input or output of size H=16, W=8464 into a squared image of size H=W=16*23.
>
> **Font size:** The font size was chosen during initial experiments with the renderer engine (both PyGame and PangoCairo) to maximise the height of the rendered characters without cropping the top or the bottom. We did not perform any experiments looking into the effects of varying the font size but this could be part of future work.
>
> **(Sub)words per patch:** The paper has some information corresponding to this part of your question. In Section 2.3, we describe the number of training examples generated for the Wikipedia and Bookscorpus dataset (“Wikipedia has 2B words rendered into 11.4M examples and the Bookcorpus has 1.1B words rendered into 5.4M examples; in total ∼3.1B words (BERT used 3.3B) rendered into 16.8M examples”). An additional analysis, not yet reported in the paper, shows that, on average, in the Bookscorpus, a patch contains 0.47 words. The efficiency analysis in Appendix D, including Figure 6, shows how many patches are needed to represent the input text across 9 languages in Universal Dependency treebanks. This analysis shows that PIXEL is at least as efficient as a discrete tokenizer (i.e. it uses the same number of patches as BERT or mBERT uses tokens). In some instances, e.g. Arabic or Japanese, the PIXEL renderer is more efficient than tokenization because it can fit more than one token in a patch.
>
> >The robustness experiments currently lack decided conclusions/takeaways. It doesn’t seem like Pixel is firmly more robust than BERT. Also, the orthographic attack graphs might need error bars?
>
> We agree that the language can be clearer and more consistent on this. In Table 5, the averages across the five results are: mBERT: 77.1, BERT: 75.8, and PIXEL: 76.5. Therefore, we conclude *on average* that PIXEL is more robust than BERT to linguistic code-switching.
>
> In order to reply to your question, we finetuned BERT and PIXEL with four additional seeds for the robustness experiments to add error bars, but, due to a technical failure, the resulting models were unexpectedly deleted from our computing infrastructure. We are rerunning the experiments and can provide the results during Discussion Stage 2. Providing multiple runs is not so straightforward for the code-switching benchmark LinCE, which uses a held-out test set and requires submitting to the leaderboard.

---

> ### Author Response · Authors · 2022-11-18
> **Reply to reviewer zqCz (2/2)**
>
> >What if Pixel is provided inputs that are rendered vastly differently during fine-tuning? It would fail to generalize right? Or would the approach be to run OCR and get the text, render it with the Pixel text renderer and then predict? It might be worth adding a discussion that goes over some of these points, and more generally delves into design decisions that are crucial.
>
> We added a new experiment in Appendix H of our revised paper in which we finetuned PIXEL models with fonts not seen during pretraining. (We note that PIXEL was only pretrained with one font: GoNotoCurrent.) We found that PIXEL is fairly robust to even substantial font variations, although performance will be best when sticking to the rendering used in pretraining. We note that generalisation across fonts or different renderings was not a primary goal for us here since we have full control over the rendering as long as we deal with standard NLP text data. Otherwise, applying font or image augmentation strategies during pretraining would likely make the model a lot more robust to changes in rendering, but we leave this study for future work.
>
> >The paper is generally comprehensive, but some details are hard to find (for instance, see previous section for suggestions). It might also be worth restructuring the results section so that it highlights conclusions a bit better – it’s a bit hard to parse right now.
>
> Thank you for the feedback! We will try to add more pointers to the appendices where many of the details can be found. We will also try to restructure the results section to make it easier to understand the main take-away messages from each experiment.

---

### Official Review · Reviewer_HZPD · 2022-10-25

**Confidence:** 4
**Correctness:** 4
**Technical Novelty And Significance:** 3
**Empirical Novelty And Significance:** 4
**Recommendation:** 8

**Clarity, Quality, Novelty And Reproducibility:**

Clarity: Extremely clear and well written+motivated! I enjoyed reading the paper.
Quality: The authors evaluated many different aspects of this model and sufficiently demonstrated its capabilities.
Novelty: To the best of my knowledge, this is the first paper to use pixel-based language systems as language models.
Reproducibility: The authors will provide a public link with access to many aspects of the model it seems.

**Strength And Weaknesses:**

Strength:
I commend the authors for the innovation here. Although a simple idea, it is highly effective and clever, and presents a very exciting alternative to BERT & friends. I look forward to future research on what these models encode and share across languages/writing systems.

Questions:
- It would be useful to have a more detailed discussion on the fertility of the pixel maps, ie., how many patches correspond to a word on average and the relationship to some of the tasks. For example, I am curious if there is a relationship between this and the question answering performance of PIXEL as the authors allude to as well.
- It would help to order the languages in the results tables based on [UNK]%. and insert a row showing the difference between BERT and pixel for each result table.
- Any hypotheses for why its performs much worse than BERT on COLA specifically?
- I understand the space and time constraints but I would be excited to see that perhaps training PIXEL on two very different script systems leads to big performance gains than traditional models and PIXEL trained on english only. Especially if the other language has a rich morphology for example.
- How hard was hyper-parameter tuning for this model? what do the loss curves look like?


**Summary Of The Paper:**

In this work, the authors developed a clever twist to the traditional neural language modeling paradigm by building a model that operates on image patches of the text. It learns to output “masked” patches using an MSE loss. Through this, the model alleviates issues with fixed or non-transferable vocabulary. When fine-tuned on a suite of downstream syntactic and semantic tasks, PIXEL observes commendable performance often on par with BERT etc. or better for languages whose writing scripts are unfamiliar to BERT. The authors also show that the model can handle code-switching and works well in the face of character-level noise.

**Summary Of The Review:**

Given the issues with vocabulary and tokenization in NLP, I think this paper offers a very innovative solution and I am curious to see how this research direction pans out.

NB:I would give this paper a 9/10 if the option existed.

---

> ### Author Response · Authors · 2022-11-18
> **Reply to reviewer HZPD**
>
> >It would be useful to have a more detailed discussion on the fertility of the pixel maps, ie., how many patches correspond to a word on average and the relationship to some of the tasks. For example, I am curious if there is a relationship between this and the question answering performance of PIXEL as the authors allude to as well.
>
> The paper has some information corresponding to your question. In Section 2.3, we describe the number of training examples generated for the Wikipedia and Bookscorpus dataset (“Wikipedia has 2B words rendered into 11.4M examples and the Bookcorpus has 1.1B words rendered into 5.4M examples; in total ∼3.1B words (BERT used 3.3B) rendered into 16.8M examples”). An additional analysis, not yet reported in the paper, shows that, on average, in the Bookscorpus, a patch contains 0.47 words.
>
> The efficiency analysis in Appendix D, including Figure 6, shows how many patches are needed to represent the input text across 9 languages in Universal Dependency treebanks. This analysis shows that PIXEL is at least as efficient as a discrete tokenizer (i.e. it uses the same number of patches as BERT or mBERT uses tokens). In some instances, e.g. Arabic or Japanese, the PIXEL renderer is more efficient than tokenization because it can fit more than one token in a patch.
>
> >Any hypotheses for why its performs much worse than BERT on COLA specifically?
>
> We have a couple of hypotheses for the larger performance gap on CoLA. First, CoLA deals with linguistic acceptability and is one of the most challenging tasks in GLUE because small nuances often decide whether the sentence is deemed acceptable or not. As such, the performance on the task tends to be lower (around 30 to 60% for most models on the GLUE leaderboard https://gluebenchmark.com/leaderboard). In such lower performance regimes, there is often a larger spread than in more saturated settings. CoLA is also one of the smallest datasets in GLUE, which means having a few more correct/incorrect predictions can have a relatively large effect on the score. Finally, in contrast to most of the other GLUE tasks, CoLA uses the Matthews correlation coefficient (MCC) performance metric, which might be more sensitive than accuracy.
>
> >I understand the space and time constraints but I would be excited to see that perhaps training PIXEL on two very different script systems leads to big performance gains than traditional models and PIXEL trained on english only. Especially if the other language has a rich morphology for example.
>
> We agree with your suggestion that such an experiment would be really interesting. Unfortunately, the mentioned space and time constraints currently hinder us from training this model. We can include a discussion of this in the future work section of the conclusion.
>
> > How hard was hyper-parameter tuning for this model? what do the loss curves look like?
>
> **Finetuning:**
> - Tables 11 and 12 in Appendix F show the range of values for hyperparameter tuning. The chosen hyperparameters tended to transfer from equivalent BERT setups.
> - As mentioned in Appendix F, tuning for GLUE was slightly more difficult, but larger batch sizes, e.g. 256, worked very well.
> - We did not do comprehensive hyperparameter sweeps due to compute limitations.
> - The loss curves decrease smoothly. More gradual / less steep decrease than for BERT. If the reviewer thinks it would be useful, we can include some representative loss curves for fine-tuning in some of the tasks.
>
> **Pretraining:**
> - Table 7 in the Appendix shows the hyperparameters that we mostly transferred from ViT-MAE’s setup. Due to compute limitations we did not conduct extensive experimentation with different variants of the details in Table 7.
> - Training was still stable with changes to dropout, masking parameters, and even batch size. We show the pretraining loss curve in Figure 7 in the Appendix.
> - Pretraining with the normalized pixel loss was crucial for downstream task performance. This was actually not clear from pretraining convergence.

---

### Author Response · Authors · 2022-11-18
**General response**

We thank the reviewers for their extensive feedback and comments about our paper. In response to each of the reviews, we have attempted to answer the questions and respond to the highlighted weaknesses. We have also updated our submission accordingly and we are happy to respond to further questions.

---

### Public Comment · ~Ada_Wan1 · 2023-04-28
**Request for submission to be retracted**

I experienced some cyber-attack/-intimidation and hence did not manage to post the following review publicly on OpenReview during the review period. I did, however, try to communicate similar information to the ICLR Board, ICLR2023 Organizing Committee, as well as the Ethics Committee before and after acceptance was announced, but did not receive any reply.

***

While it's great to see further development in the direction of visual text representation (Salesky et al. (2021)), that one can use pixels to model texts, I remain apprehensive of the authors' choice of evaluation methods.

The word-level tasks involve "word" tokenization. Would there not be a difference in performance if a different tokenizer were used? One result from Wan (2022) is to point out the problem of vocabulary hacking in NLP along with the indeterminacy of "words" and its resolution into more standardized/stable units. Using a fuzzy term such as "word" can surely further such abuse. Also as seen in Appendix F, there is information loss that results from converting text representation to words. This paper evaluates on different amounts of data. Recall representational relativity: "language has many finer-grained dimensions with different representations and learning patterns. Hardness in modeling is relative to its representational granularity (representation relativity)" (Wan, 2022).

Therefore, an explicit analysis/evaluation of this process is necessary BEFORE this present work should be published. The acceptance of this work needs to be retracted.

Any task that is grammatical in nature (syntactic, semantic etc. based on "words") is not and should not be an accurate or sufficient method for text evaluation in the computational setting.

Wan (2022) was not so much about vocabulary "construction" but an investigative evaluation thereof rather. One (other/better) place where the work could/should have been cited is along where the authors mentioned "vocabularies over bytes or characters are much smaller, which leads to increased sequence lengths" on p.1.. One other important citation missing is that of Mielke et al. (2019). Both of these work conclude that data statistics are the reasons behind performance disparity in multilingual text representations. Have authors examined to which extent data statistics play a role in the study? Has one reconciled crosslingual fairness issues with data statistics and other statistical conditions? Reporting of statistical significance and/or mean and variance information would also be helpful.

As technologists, one does/should have the capacity to make things fair(er) in the multilingual front in computing (see discussions for Wan (2022)). Authors could have made things fairer through, e.g. alternate data representation strategies or hyperparameter tuning.

Additional missing citation (re vocabulary bottleneck with softmax): Yang et al. (2018).


References:

Ada Wan. Fairness in representation for multilingual NLP: Insights from controlled experiments on conditional language modeling. In International Conference on Learning Representations, 2022. (https://openreview.net/forum?id=-llS6TiOew) [See discussions and slides also.]

Sabrina J. Mielke, Ryan Cotterell, Kyle Gorman, Brian Roark, and Jason Eisner. What kind of language is hard to language-model? In Proceedings of the 57th Annual Meeting of the Association for Computational Linguistics, pp. 4975–4989, Florence, Italy, July 2019. Association for Computational Linguistics. doi: 10.18653/v1/P19-1491. (https://arxiv.org/pdf/1906.04726.pdf).

Zhilin Yang, Zihang Dai, Ruslan Salakhutdinov, William W. Cohen. Breaking the Softmax Bottleneck: A High-Rank RNN Language Model. In International Conference on Learning Representations, 2018. (https://openreview.net/pdf?id=HkwZSG-CZ).

---

### Decision · Program_Chairs · 2023-01-20

**Decision:**

Accept: notable-top-5%

**Justification For Why Not Higher Score:**

Already highest. I think this is an interesting paper deserving of an oral which would gain alot of attention (as it deserves)

**Justification For Why Not Lower Score:**

This paper is highly innovative and interesting and could be pretty impactful down the line.

**Metareview: Summary, Strengths And Weaknesses:**

This paper presents a novel approach of learning LMs from pixels and is essentially a vocab-free method for language.

All reviewers liked the paper and think this approach is highly novel and interesting.

I strongly recommend acceptance.

The AC likes the paper too.

**Note From Pc:**

if the above contains the word "oral" or "spotlight" please see: "oral" presentation means -> notable-top-5% and "spotlight" means -> notable-top-25%. As stated in our emails, we are disassociating presentation type from AC recommendations